# Two is better than one: Using a single emotion lexicon can lead to unreliable conclusions

**Gabriela Czarnek** [1], **David Stillwell**[2,3]*

1 Institute of Psychology, Jagiellonian University, Krakow, Poland, 2 Psychometrics Centre, University of Cambridge, Cambridge, United Kingdom, 3 Judge Business School, University of Cambridge, Cambridge, United Kingdom

* d.stillwell@jbs.cam.ac.uk

## Abstract

Emotion lexicons became a popular method for quantifying affect in large amounts of textual data (e.g., social media posts). There are multiple independently developed emotion lexicons which tend to correlate positively with one another but not entirely. Such differences between lexicons may not matter if they are just unsystematic noise, but if there are systematic differences this could affect conclusions of a study. The goal of this paper is to examine whether two extensively used, apparently domain-independent lexicons for emotion analysis would give the same answer to a theory-driven research question. Specifically, we use the Linguistic Inquiry and Word Count (LIWC) and NRC Word-Emotion Association Lexicon (NRC). As an example, we investigate whether older people have more positive expression through their language use. We examined nearly 5 million tweets created by 3,573 people between 18 to 78 years old and found that both methods show an increase in positive affect until age 50. After that age, however, according to LIWC, positive affect drops sharply, whereas according to NRC, the growth of positive affect increases steadily until age 65 and then levels off. Thus, using one or the other method would lead researchers to drastically different theoretical conclusions regarding affect in older age. We unpack why the two methods give inconsistent conclusions and show this was mostly due to a particular class of words: those related to politics. We conclude that using a single lexicon might lead to unreliable conclusions, so we suggest that researchers should routinely use at least two lexicons. If both lexicons come to the same conclusion then the research evidence is reliable, but if not then researchers should further examine the lexicons to find out what difference might be causing inconclusive result.

**Data Availability Statement:** To protect users privacy and abide by Twitter's policy we can only share tweet IDs. We share age/birthday announcement tweets IDs that were used to create our dataset at https://osf.io/pn5f6/.

## Introduction

### Goals

Today textual data is created and collected at an unprecedented scale, in real-time, and is oftentimes freely available. Due to its volume, natural language processing is used to gain insights from such data that comes from, e.g., social media, news, or political speeches [1, 2].

**Funding:** This research was supported by the Ministry of Science and Higher Education in Poland with a MOBILITY PLUS grant (no 1614/MOB/V/2017/0) received by Gabriela Czarnek. The funders had no role in study design, data collection and analysis, decision to publish, or preparation of the manuscript.

**Competing interests:** The authors have declared that no competing interests exist.

One seemingly straightforward and frequently employed method are lexicons which have been developed to quantitatively analyse categories of expression [3]. In this paper, we focus specifically on emotion lexicons.

Researchers can often choose from multiple independently developed emotion lexicons and it might be challenging to select just one lexicon for a their analysis. Fortunately, a number of studies have compared different lexicons measuring the same emotions with the goal of finding the best one to use. To do this, typically researchers correlate the scores from different lexicons or correlate lexicon scores against a criterion such as human ratings of emotional expression in a particular text [4–7]. It is useful to identify the most reliable lexicon (especially if it is to be applied to a small sample of text where errors may not average out). But more concerning for the applied researchers is the possibility that lexicons may be systematically different from one another. That is, the measure works differently in different contexts (e.g., tweets versus Facebook status updates) or for different subgroups of people (e.g., men versus women). However, most previous research comparing lexicons did not verify whether testing a theory-driven hypothesis with different emotion lexicons leads to the same conclusions. In other words, are the relationships between lexicon scores and external variables comparable? Thus, in this study, we examine what would be the conclusions of a study testing a single research question using two different and widely used emotion lexicons.

As an illustrative example of a theory that is testable using emotion lexicons, we examine theories of age-related changes in emotional experience: Do older people express more [8] or less [9] positive affect as they get past retirement age? Specifically, we investigate patterns of emotional expression in tweets among users (N = 3,573) from a broad age range (between 18–78 years old). To quantify emotional expressions, we employ two popular emotion lexicons: Linguistic Inquiry and Word Count software (LIWC) [10] and the NRC Emotion Lexicon (NRC) [11, 12]. Would the two methods give the same answer to the research question regarding emotional expression in older age?

## Emotion lexicons

Emotion lexicons are lists of words, sometimes short phrases or emoticons, which are categorized as reflecting particular emotional expressions, e.g., positive affect or surprise. It is worth noting that in psychology, the terms *affect* and *emotions* are not interchangeable: affect can be positive, negative, or neutral and experienced as free-floating (mood) or a short-lived state; whereas specific emotion, e.g., anger or contempt, is a complex pattern of physiological (e.g., hormones levels), cognitive (e.g., interpretations), and behavioural changes (e.g., facial expressions) in response to a particular object or situation [13, 14]. In a similar vein, some researchers use the term "sentiment analysis", which comes from early work on detecting subjectivity of customer reviews [15]; it refers to extracting polarity from texts (positive, negative, or neutral, similar to the notion of affect). Others differentiate it from "emotion analysis" which focuses on understanding specific emotions e.g., [16]. We use the terms "emotion lexicon" broadly, i.e., any list of words that can capture expression of feelings, affect, mood, sentiments, or specific emotion, similarly to [15].

Many lexicons are available, often measuring exactly the same emotions but differing by the number of terms they contain, the method of their creation, or the goals with which they were created. With regard to method of creation, lexicons could be divided into two broad categories: human-labelled vs. machine-learned.

Human-labelled methods involve human annotators, experts or crowd-sourced, who provide an evaluation of emotional qualities of words which later become lexicon terms [17]. Experts are usually just a few psychology or linguistics professionals. Typically, they

individually decide whether candidate words are an acceptable example of a certain emotional expression, and later discuss the inconsistencies between their ratings, and agree on a final lists of lexicon words, which is also supported by the psychometric analysis. In contrast, crowd-sourcing involves large number of lay people (e.g., MTurk workers, [11]) who provide ratings of several words (and not all the words); their rating are later aggregated and the statistical agreement determines the lexicon words.

Additionally, some lexicons provide lists of words that *denote* emotions, that is words that express a particular emotion irrespective of the context, e.g., "rage", "great" [15]. Other lexicons instead, are broader in the approach and focus on words' emotional *connotation*, i.e., whether the words are associated with particular emotion, e.g., "loss", "friendship" [15]. Importantly, words that denote emotions (e.g.,"great") are also associated with the emotion, thus might be included in such a lexicon. The denotative lexicons have the advantage that the instances of false positives (e.g. a word incorrectly labelled as positive which is actually not) should be low, but connotative lexicons have more scope to identify a larger list of words so the instances of false negatives (e.g. a word incorrectly *not* labelled as positive which actually is) should be low. Nevertheless, because involving people, experts or non-experts, is rather costly and lengthy, these lexicons are usually not longer than a few hundred to thousand words [15]. In consequence, such lexicons might have low coverage of the total number of words in a text. Thus, automated are employed to creating word-association lexicons.

Creating a lexicon with machine-learning typically involve a labelled dataset of sentences that express a certain emotion (training dataset); various techniques or their combination can be used to identify a list of words which correlate with the ground truth labelled data [17]. For example, researchers created a large dataset with tweets containing one of nearly 80 emotion-word hashtags; if a tweet had a hashtag denoting positive emotion (e.g., #great), it was considered being positive, and if it had a hashtag denoting a negative emotion (e.g., #terrible), it was considered being negative. Next, they used a machine-learning classifier on this training data to create a lexicon containing entries for ~54 thousand unigrams (single words or terms) and ~317 thousand bigram (two-term sequences). Importantly, machine-learned lexicons provide word-emotion associations, i.e., they are connotative. Machine learning methods can be used to create lexicons that are suited to a particular domain given that words might have different emotional connotations given the context, e.g., [18] as opposed to more general human-labelled lexicons. For more comprehensive and non-technical overview of methods of automatic detection of emotions in texts, including emotion lexicons, see [15].

In this paper, we focus on emotional expressions captured by lexicons that have been created using human-labelling: they are easy to use and it is straightforward to assess which particular words contribute to the overall affect scores. These advantages made human-labelled lexicons widely used in social sciences which are focused on testing and developing theories, as opposed to computer science research which might focus more on creating the most accurate algorithm to recognize emotional qualities in texts regardless of the complexity or transparency of the algorithm.

## Lexicons in the current study

As already mentioned, in the current study, we use two popular lexicons, LIWC [10] and the NRC Emotion Lexicon (NRC) [11, 12]. We provide a detailed description of the two lexicons below.

The 2015 version of LIWC [10] has nearly 6,400 terms within 90 psychologically meaningful categories of language, e.g., linguistic features such as first-person pronouns, cognition- or health-related terms. Out of these wide-ranging 90 categories, seven provide emotional

wordlists: "affective processes" (1,393 emotional terms irrespective of their valence, e.g., "happy", "cried"), "positive emotion" (620 terms, e.g., "love", "nice"), "negative emotion" (744 terms, e.g., "hurt", "ugly"), "anxiety" (116 terms, e.g., "worried", "terrify"), "anger" (230 terms, e.g., "annoy", "hate"), and "sadness" (136 terms, e.g., "grief", "cry"). The categories in LIWC are organized hierarchically: the terms in specific emotions will be included in the positive/negative emotion lists, and all of them will be in "affective processes" category. Importantly, in this paper we focus on positive and negative affect (and not on specific emotion); "positive emotion" and "negative emotion" categories have 1,364 terms in total. Some of the LIWC terms are stemmed words which expand the lexicon significantly: all the terms sharing a particular stem are recognized and counted as a term occurrence. For example, for the word stem "happi∗" all the words starting from "happi" such as "happiness", "happier", and "happiest" will be counted (but the word "happy" will not).

Each category of LIWC, including the emotional lexicons, was created in several steps. During *Word Collection*, the emotional words were harvested from existing psychological scales (e.g., words such as upset or proud from PANAS [14]), their synonyms from a thesaurus were added; next a group of several judges generated new words individually and later in a group brainstorming sessions. Next, in the *Judge Rating Phase*, the collected wordlists were qualitatively evaluated by a panel of the judges. For a word to be included in a particular category, the majority of the experts needed to agree on its goodness-of-fit to that category; if they could not decide on the appropriate category, a word was removed from the lexicon. In the *Base Rate Analysis*, the frequencies of the words were evaluated using multiple sources of text data (e.g., blog posts, Twitter, spoken language); if the word did not occur at least once in several of those texts, it was removed from the lexicon. Next, in *Candidate Word List Generation*, new words were harvested from multiple texts from previous studies; if a word had high frequency, was not included in LIWC already, and was correlating with a LIWC category, several judges decided on suitability for inclusion of that word. In the *Psychometric Evaluation*, a psychometric analysis of each category was conducted; if a word was detrimental to a category's internal consistency it was flagged, and the panel of judges again, decided on whether to keep it or remove it from the lexicon. All these steps were repeated in order to refine the lexicon and catch potential mistakes; however, the Authors note that the changes in this *Refinement Phase* were marginal. Overall, the process of creating the lexicon is largely based on the expert consensus but also seem time-consuming and rather high in costs.

LIWC is available with an accompanying software for a moderate fee. Using LIWC does not require any programming skills so it is accessible to a broad range of users. The new, fifth, LIWC version has been released in February 2022 [19]; and for new users, it is only possible to purchase fifth version through the LIWC app (through which 2015 and earlier LIWC versions can be accessed). The major differences between 2015 and the new 2022 LIWC version are: adding new categories (e.g., "ethnicity", "fatigue"), expanding wordlists in the existing categories (it now has over 12,000 terms in total), and removing a few categories due to their low base rates (e.g., "comparison words"). For the emotion lexicons, the important changes include: replacing "positive emotion" and "negative emotion" categories with "positive sentiment" and "negative sentiment", respectively (however, the respective lists are very similar and the correlations between 2015 and 2022 scores for a sample text is around 0.85 [19]); changing the content of specific emotion lexicons so that they now include only denotative words or strongly associated words; and excluding the swear words from positive and negative sentiment/emotion categories; the details are available in the 2022 LIWC psychometric manual [19]. The conclusions from the current paper, which uses the 2015 LIWC version, we believe, are still important and illustrative of a larger issue related to the usage of emotion lexicons, which just does not pertain to a particular lexicon or its version; we elaborate on this issue in

the Discussion section. Furthermore, given that LIWC 2022 has just been released, we expect many researchers still use the 2015 LIWC version.

LIWC has been used in research programs covering a wide range of topics including attitudes [20], mental disorders [21, 22], individual [23, 24] and gender differences [25, 26], as well as group [27] and collective processes [28, 29]. Due to its comprehensiveness and user-friendly software, LIWC became the main lexicon for research in psychology and related disciplines, currently being cited more than 20 thousand times according to Google Scholar. It is, however, important for the accuracy of the automated affect assessment that a large proportion of words from a text are covered by the lexicon's terms [5]. Thus, researchers might prefer to use lexicons providing longer wordlists.

In contrast to LIWC, The NRC Emotion Lexicon [11, 12] focuses on emotional words only, and provides a list of 13,872 terms in several categories. Specifically, the terms are classified as "negative sentiment" (3,316 terms, e.g., "homeless", "yell"), "positive sentiment" (2,308 terms, e.g., "optimist", "youth"), and eight specific emotions, based on Plutchik's theory [30]: "anger" (1,245 terms, e.g., "aggression", "soldier"), "disgust" (1,056 terms, e.g., "pus", "butcher"), "fear" (1,474 terms, e.g., "shady", "lion"), "sadness" (1,187 terms, e.g., "grieve", "scarce"), "joy" (667 terms, e.g., "praise", "picnic"), "anticipation" (837 terms, e.g., "foresee", "vow"), "trust" (1,230 terms, e.g., "witness", "sex"), and "surprise" (532 terms, e.g., "incident", "birthday"). The "positive sentiment" and "negative sentiment" wordlists that we will focus in this paper, have 5,624 terms in total. NRC is freely available for academic use at [31] but requires some coding skills in order to use it, thus might be less popular researchers from social sciences and humanities. Recently, the official Python package was released for analysing texts with NRC [32].

The initial set of NRC candidate terms was selected from other emotion lexicons (General Inquirer [33], WordNet Affect Lexicon [34]) and from Google n-gram [35] (words with the highest frequency), resulting in a pool of around 10 thousand terms. Next, each term was evaluated by a few (4.4 on average) Amazon Mechanical Turk workers who decided how positive and negative a word is as well as how strongly it is associated with each of the eight emotions. Importantly, the raters were presented with the synonyms of a word so that evaluation was provided on a *sense* rather than a *word* level [12]. For example, when the word "startle" was evaluated, MTurkers were nudged with the word "shake" as its synonym and the same word could have been evaluated with several different meanings. The evaluation for sentiments was provided on a 4-point scale: "not (positive)", "weakly (positive)", "moderately (positive)", and "strongly (positive)". It is also important to note that due to such an approach, NRC provides word-emotion associations, i.e., it is a connotative lexicon. After screening the annotations (e.g., for mistakes or outliers), nearly nine thousand terms remained in the pool; these terms were used to create the final lexicon. The classification of each term into categories was determined by the intensity level that was chosen most often by raters (the majority vote).The first two categories ("not (positive)", "weakly (positive)") were treated as not associated with a particular affect, and the last two categories ("moderately (positive)", "strongly (positive)") were treated as associated with a particular affect. Ratings for words which had more than one sense were collapsed into one category and the majority vote served as its final category. Because each term could be associated into multiple categories, the resulting lexicon has around 14 thousand entries.

The NRC Emotion Lexicon, was the first word-association lexicon that provides comprehensive lists of two sentiments and eight emotions; it is also still the largest such a list [31]. It was also one of the first lexicons that employed a large group of crowdsourced annotators, as opposed to a small group of competent judges. From its creation more than 10 years ago, NRC became widely popular; it has been used, for example, in a study of communication around infectious diseases [36], detection of hate speech [37], or identifying early signals of the

financial crisis in bank annual reports [38]. Importantly, it has also been used as a seed lexicon for many advanced machine-learning applications e.g., [39].

Although both LIWC and NRC use human-labelling, their scope and methodological approach are certainly different. To recap, LIWC categories were created by an iterative process of qualitative analysis, experts' discussions on the classification of words, and the analysis of psychometric properties of the lexicon terms. In contrast, NRC's terms were rated by non-expert annotators recruited online, and the majority vote determined a category of a word. Importantly, although LIWC seem to be including both words expressing emotions (denotation), it includes also emotion-word associations that the judges were able to agree upon. In contrast, NRC focuses on word-emotion associations, i.e., it includes terms expressing particular emotions but the associations between a term and an emotion might be more or less close given that both "strong" and "moderate" associations were accepted. Furthermore, although LIWC became commonly used in social sciences and humanities, NRC provide a much longer list of emotional words. For example, for positive and negative affect LIWC provides around 1.3 thousand terms (including word stems) and NRC provides around 5.6 thousand terms. This makes NRC potentially very useful for analysis of a wide range of texts, especially short ones where LIWC may not match enough affective words for a reliable estimate.

Given these differences between LIWC and NRC, we compare what conclusions for psychological theory one would make using one or the other lexicon on the same dataset. We focus on the relationship between age and emotional expressions as an example of a seemingly simple question that is still actively debated by researchers.

## Emotional experience across age

We test whether the LIWC and NRC lexicons lead to differing conclusions in the theoretical context of changes in emotional experience across age. In contrast to lay beliefs [40, 41], the psychological literature shows that older age is not necessarily related to the feelings of sadness, loneliness, and loss. Despite factors usually predicting low levels of well-being, including compromised health, declining cognitive abilities, and shrinking social networks, emotional experience is actually maintained or even increased in older age [42–44]. This effect is known as the *paradox of ageing*.

One of the influential accounts of this effect is Socio-emotional Selectivity Theory [8, 43, 45] which posits that as people get older, due to an increasing perception of limited time to live, they become more focused on the regulation of their emotions and prioritize well-being over other competing goals. Older adults, thus, focus on increasing positive affect and minimizing negative affect by selectively engaging in emotionally gratifying activities and avoiding unpleasantness. For example, older adults may withdraw from maintaining numerous superficial relationships and instead intensify contacts with people with whom they have meaningful, close rapport. In other words, the mechanism of increased well-being is goal selection, or self-regulation more broadly. Some research suggests that age-related increase in well-being is driven by a concurrent increase in positive and decrease in negative affect [46, 47]. Nevertheless, other studies have shown that changes in well-being occur due to a decrease in negative affect with positive affect not changing in older age [48, 49]. Thus, it is also important to separately investigate expressions of positive and negative affect as opposed to solely relying on the overall affective experience.

We investigate whether using two popular emotion lexicons, LIWC and NRC, gives consistent answers to a psychological theory-driven research question: do people experience more positive affect and less negative affect as they grow old? Our study, comprising several thousand participants, allows for tracking everyday spontaneous emotional expressions and testing the competing theories of age-related changes in emotional well-being.

## Materials and methods

### Data collection

Data was collected from Twitter by searching for users who publicly disclosed their age in a tweet, e.g. posting "I'm 50 today, happy birthday to me!". We aimed to identify at least 40 age announcement tweets for each age between 18 and 80 (which gives around 2,500 age announcement tweets in total). Several research assistants collected tweets between 28[th] January and 26[th] February 2019 and provided 5,124 such tweets. After initial filtering of the users (the details are in the Data cleaning section), we downloaded users' timelines with all available tweets on the 31[st] March 2019; up to 3,200 tweets could be downloaded from the Twitter API for each user. The age announcement tweet and its timestamp allowed us to identify users' date of birth and we used this to calculate the user's age at the time of each tweet. For example, if a user announced on 17[th] June 2018 that she is 30 years old, we treated all her tweets posted between 17[th] June 2018 and 16[th] June 2019 as those of a user who is 30 years old, tweets posted between 17[th] June 2017 and 16[th] June 2018 as of a user who is 29 years old and so on. Thus, in our dataset, a document is a collection of tweets for a particular user at a particular age. Data was collected and cleaned with R [50] and Rstudio [51] with twitter [52], rtweet [53], readxl [54], tidyr [55], dplyr [56], tidytext [57], and stringr [58] packages. This research underwent ethical review by Cambridge University Judge Business School's IRB and was in accordance with the Twitter terms of service.

### Data cleaning

We cleaned Twitter timelines both on a user- and a tweet-level. Starting with the user-level, we were able to retrieve age announcement tweets of 4,826 users out of 5,124 tweet links harvested by research assistants (some tweets were removed, accounts became private, etc.). Next, we included only public Twitter accounts where users specified English as their language, which left 4,754 users in the dataset. To remove celebrities and accounts which might be curated by PR companies we excluded verified accounts or those users who had more than 2000 followers, e.g., as [59, 60]; that left 3,998 users. We have also removed accounts that had less than 20 tweets in total; this left 3,938 users in the dataset. We also excluded accounts which were likely to be automated, i.e., bots, by setting a cut-off of 0.5 in botometer score [61]. Botometer checks many aspects of account activity, e.g., users' networks or timing patterns of tweeting, and provides a score between 0–1 on how likely a user is a bot. The higher the score, the more likely an account is bot-like. We accessed the botometer API through the botcheck package in R [62]. This has left 3,642 users in the dataset.

On the tweet-level, we retained only original posts and excluded 2.5 million re-tweets, which left around 6 million tweets in the dataset. Despite including only English Twitter accounts, some tweets turned out not to be English. To exclude them, we passed all the tweets from the dataset to Google's Compact Language Detector implemented in R in the cld2 and cld3 packages [63, 64]. If both algorithms detected a language other than English, a tweet was removed (around 1.1 million tweets were removed and left nearly 5 million tweets in the dataset). Next, within each tweet, we removed all Twitter users' mentions, emojis, hashtags, and characters such as ampersands and *greater/less than* signs. All remaining words were lower-cased. Furthermore, to ensure an adequate level of reliability, i.e., having enough words to analyse emotional expressions, we also excluded observations which had fewer than 20 tweets per document (excluding only those years in which there were fewer than 20 original tweets but retaining years for which there were at least 20 tweets), which have left 3,600 users in the dataset. Finally, we decided to exclude documents if users' age is below 18 years old and those

older than 78, due to ethical reasons and low frequencies, respectively; thus in the final dataset 3,573 users remained.

After data cleaning, 71,312,332 terms in 4,878,690 tweets written by 3,573 users remained in the dataset. The average age of the users was 43.58 (SD = 15.91). For each user, we had at least one document (tweets written while a certain age) and a maximum of 11 documents, with the majority of users having between one and three documents (the average was 3.40, SD = 2.13) for a total of 12,162 documents. A histogram depicting distribution of age in the sample is shown in S1 Fig of S1 File.

## Measures

As already mentioned, we analysed positive and negative affect using LIWC and NRC. We determined LIWC positive and negative affect scores using dedicated software [10]. For the NRC Emotion Lexicon [11, 12], we used a lexicon implemented in the tidytext R package [57]. To calculate affect scores, we counted the occurrence of each word and divided it by the total number of words per user in a particular year (i.e., a document). The resulting scores represent a percentage of emotional words out of a total number of words. After obtaining scores for positive and negative affect, we computed overall affect by subtracting negative from positive affect scores, separately for LIWC and NRC. Finally, we removed scores which deviated more than +/- 3 SD from the average score for any measure.

## Results

To analyse the age-related change in emotional expressions we ran a series of multilevel models in which we nested documents (ages) within participants (random-intercept model). We fitted autoregressive models with linear and quadratic effects of grand-mean centered age. Furthermore, due to small values of the coefficients for 1 year as a unit, for easier interpretation we use 10 years as a unit; i.e., the interpretation of the age coefficient is "an increase in 10 years is associated with an X change in emotional expression". Data was analysed with R [50] and Rstudio [51] with nlme [65], effects [66], ggplot2 [67], ggwordcloud [68], and texreg [69] packages.

## Original scores

**Overall affect.** For the overall affect, the models with both linear and quadratic age effects outperformed the models with only linear effects of age both for LIWC and NRC. The model comparisons are presented in S1 Table of S1 File and the model details are shown in S2 Table of S1 File. According to both the LIWC and NRC models' predictions, shown in the first columns of Fig 1 (first row presents LIWC scores, second row NRC scores), the overall affect increases up until age 50 and then decreases. However, LIWC reports that the inverted U-shaped relationship between age and overall affect is symmetric: the youngest and the oldest people have the lowest overall affect scores. But NRC reports that although this overall affect relationship is an inverted U-shape, it is not symmetric: people in their early 20s have lower scores than those in their late 70s. Nevertheless, the differences in LIWC and NRC predictions for the overall affect are not very large.

**Positive affect.** For positive affect, the models with linear and quadratic effects of age also outperformed the simpler ones for both LIWC and NRC (the model comparisons and details are presented in S1, S2 Tables of S1 File, respectively). However, the model predictions for LIWC and NRC, shown in the middle columns of Fig 1 (LIWC in the first row, NRC in the second one), differ to a rather large extent in the part of the curve that is critical for our theoretical research question. Both models predicted an increase in positive affect until age 50.

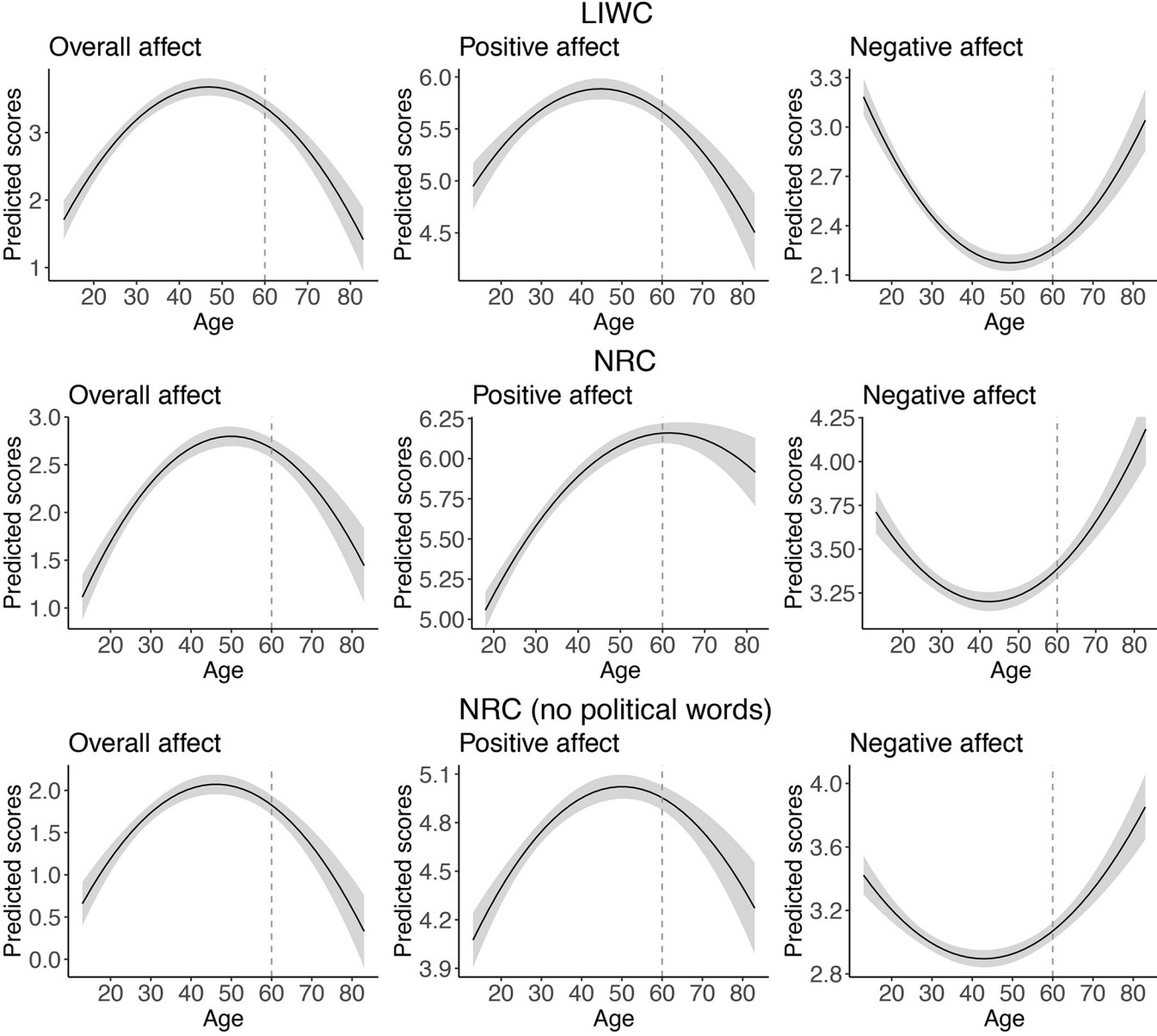

**Fig 1. Predicted scores of the overall, positive, and negative affect scores.** First row represents model predictions based on LIWC scores. Second row represents model prediction based on NRC scores (full lexicon). Third row shows model predictions based on re-calculated NRC scores(after removing political words).

After that age, however, according to LIWC, positive affect drops sharply, whereas according to NRC, the growth of positive affect increases steadily until age 65 and then just levels off. While the NRC scores are in line with socio-emotional ageing theories, the findings from LIWC are not consistent with them.

**Negative affect.** For negative affect, the models with linear and quadratic effects of age also outperformed the simpler ones for both LIWC and NRC (the model comparisons and details are presented in S1 and S2 Tables of S1 File, respectively). However, the model predictions for LIWC and NRC, shown in the third column of Fig 1 (LIWC in the first row, NRC in the second one), differ somewhat. While both models predict a U-shaped relationship between

age and negative affect, with the lowest negative affect scores for people between 40 to 50 years old, they differ in the predictions for youngest and oldest groups. According to the LIWC model, we should expect the highest negative affect scores for young people (those below 25 years old), whereas the NRC model predicts the highest negative affect scores should be expected for older people (those above 70 years old). Neither of the lexicons supports the premises of the ageing theories, which suggest a decrease in negative affect as people grow older.

## Words' contributions

According to the analysis of the relationship between age and emotion expression, it seems that depending on which lexicon is used, NRC or LIWC, researchers might have come to different conclusions about the relationship between age and affect. NRC largely upheld the predictions from Socio-emotional Selectivity Theory for the positive affect, whereas LIWC would have rejected them. Thus, next we will examine the LIWC and NRC lexicons in more detail to understand why their predictions differ, especially for the positive affect scores.

First, we evaluate which lexicon terms had a high contribution to the affect scores across LIWC and NRC. We calculated Pearson's r correlations between age and all LIWC as well as NRC terms. We took into consideration only unique, non-overlapping terms in the two lexicons: there were 700 unique LIWC terms (367 positive and 333 negative affect terms), and 4,945 unique NRC terms (2,038 positive and 2,907 negative affect terms). Because LIWC provides not only words but also word stems, we matched the stems by filling them with available NRC terms. If there were several NRC words that would match a LIWC stem, we used only the first one to complete a LIWC word stem. For LIWC, there were 274 unique terms significantly correlated with age (140 from the positive and 134 from the negative affect wordlist). For NRC, there were 1,837 unique terms significantly correlated with age (755 from the positive and 1,082 from the negative affect wordlist).

The word clouds for positive and negative affect words are presented in Figs 2 and 3, respectively. For presentation purposes, we have displayed only those terms where the absolute value of correlation coefficient was equal to or higher than 0.05. While the size of a word corresponds to its average frequency, the colour corresponds to the direction and intensity of the correlation with age: the more vivid blue a word is, the more it is related to a young age; the more vivid red a word is, the more it is related to an older age.

In terms of the number of words, unsurprisingly, as there are fewer unique LIWC terms, a smaller number of them were correlated with age when compared to NRC. It is quite evident when the LIWC word clouds are compared against NRC word clouds. For unique LIWC words, it seems that a similar number of terms correlate with young and older age for positive and negative affect. In contrast, for NRC, there are more words correlated with older than younger age both for positive and negative affect.

In terms of content, the word clouds for terms related to younger age, both NRC and LIWC, seem to have face validity, and so does the LIWC word clouds for terms related to older age for both positive and negative affect. For example, both LIWC and NRC word clouds show that younger age is related to words such as "lol", "besties", "hangover", "weirdos" (LIWC) and "baby', "cuddle", "ass", "boo" (NRC), whereas older age is related to words such as "care", "support", "vile, "pity" (LIWC). However, in the NRC word clouds for older age several terms seem to be related to politics, both for the positive and negative affect, e.g., for the positive NRC wordlist: "president", "police", "cabinet"; for the negative NRC wordlist: "government", "foreign", "liberal"; and surprisingly several words are crossed-classified as positive and negative affect, e.g., "vote", a word with a high frequency. We found 81 NRC terms classified as

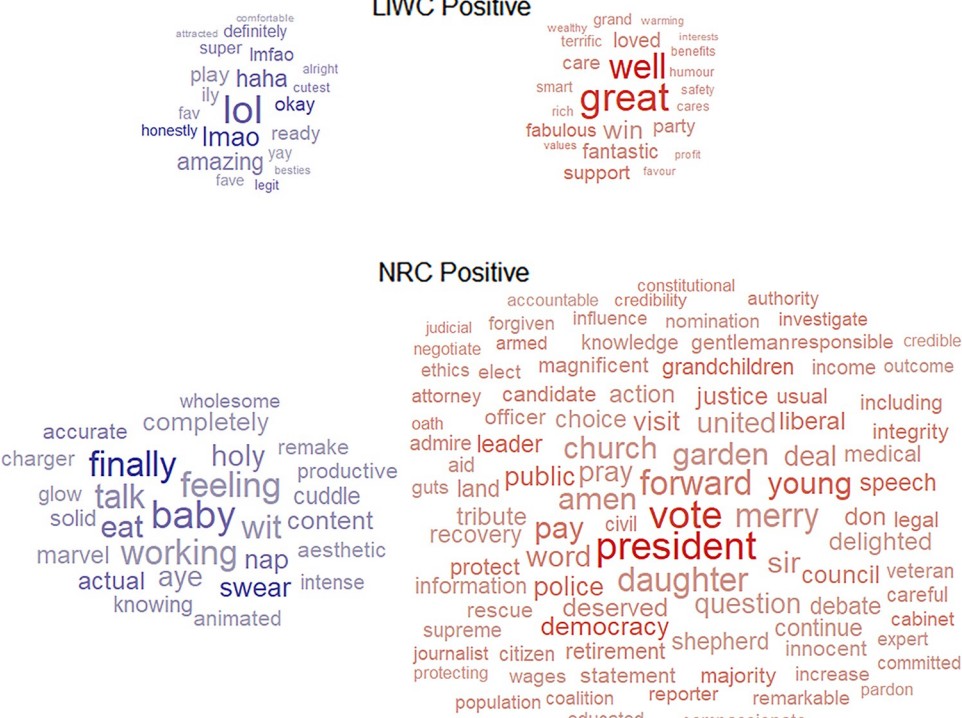

**Fig 2. Unique positive affect words from LIWC and NRC that are correlated with age.** Size of a word corresponds to its average frequency in the dataset, the colour corresponds to the correlation with older age: the more blue, the more negative correlation; the more red, the more positive correlation.

both positive *and* negative, e.g., "weight", "vote", "teens", "mug", "income", "midwife", "hedonism". These cross-classified words will be removed from the subsequent analyses. More importantly, however, because the political words were not present in LIWC word clouds, they are a candidate class of words that could be responsible for the inconsistency in NRC and LIWC scores for people in older age. In the next section we will identify the political words that could be causing these inconsistencies.

## The influence of the political words

Given words' contribution to the emotion scores presented above, our working hypothesis is that the political words in the NRC could be responsible for the differences between the NRC and LIWC scores among older people. In order to verify whether this is the case, we aimed at excluding political words which might contribute to the differences between NRC and LIWC model predictions. The logic of this analysis is the following: we aimed to identify groups of words that are related to politics and frequently used by people in older age (because only those seem to be creating a difference between NRC and LIWC scores); then, we try to select the subgroup of these words that could be causing the inconsistency between NRC and LIWC by correlating the frequency of using these words with the LIWC emotion scores.

We started our analysis by identifying political words using Latent Dirichlet Allocation (LDA) [70], a topic modelling technique that produces sets of words that tend to co-occur in a document. We first excluded stopwords, i.e., frequent but not informative words (e.g., "we", "the" [57]) and 27,414,128 terms remained for the LDA analysis. After identifying the appropriate number of topics using several criteria [71–74] with the ldatuning package [75] (see the

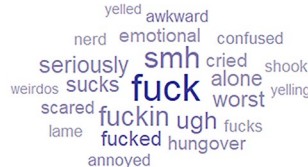
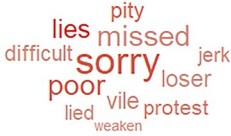
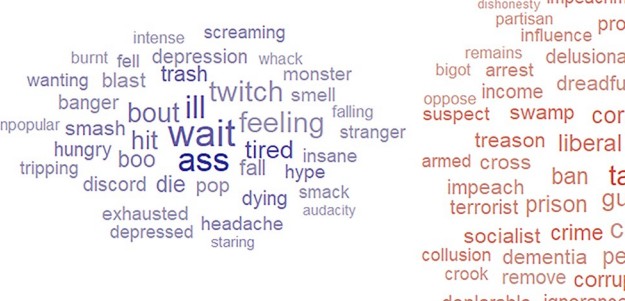

**Fig 3. Unique negative affect words from LIWC and NRC emotion lexicon that are correlated with age.** Size of a word corresponds to its average frequency in the dataset, the colour corresponds to the correlation with older age: the more blue a word the more negative correlation; the more red a word the more positive correlation.

S3 Fig in S1 File for details), we modelled 50 topics with the topicmodels package [76]. However, including all of the 50 topics in the subsequent analyses could lead to over-fitting, multi-collinearity, and thus problems with a model interpretation. Hence, we decided to trim the number of topics using penalized regression (LASSO regression). Specifically, we extracted topic probabilities for each document, that is the probability of a topic given a user (at a particular age). However, we chose just one document per user (the most recent one) because the documents within the same person are correlated and LASSO regression did not allow us to account for such dependencies in the data. We entered users' age as a dependent variable and 50 topic probabilities as predictors in the LASSO regression using glmnet package [77] (the value of lambda parameter was determined using cross-validation). The LASSO regression analysis revealed 36 topics with non-zero values of the coefficients and only these topics were included in the subsequent analysis.

Because LASSO regression did not allow us to use all available data (we could not properly account for a multilevel structure of our dataset), we next used an analytic technique that allows for a more comprehensive analysis of what topics are more frequently used in older age. Specifically, we ran a multilevel regression analysis with all available documents (documents were nested within users by including per-user random intercept). Again, users' age was predicted from the selected 36 topics probabilities (the predictors were standardized). This analysis revealed 22 topics correlated with age. The details of this model are presented in S3 Table of S1 File; the word clouds presenting the topics' top words is available in S4 Fig of S1 File. Out of 22 topics correlated with age, there were four topics relevant to politics; the top 15 terms from these four political topics are presented in Fig 4. The first topic seems to reflect politics in

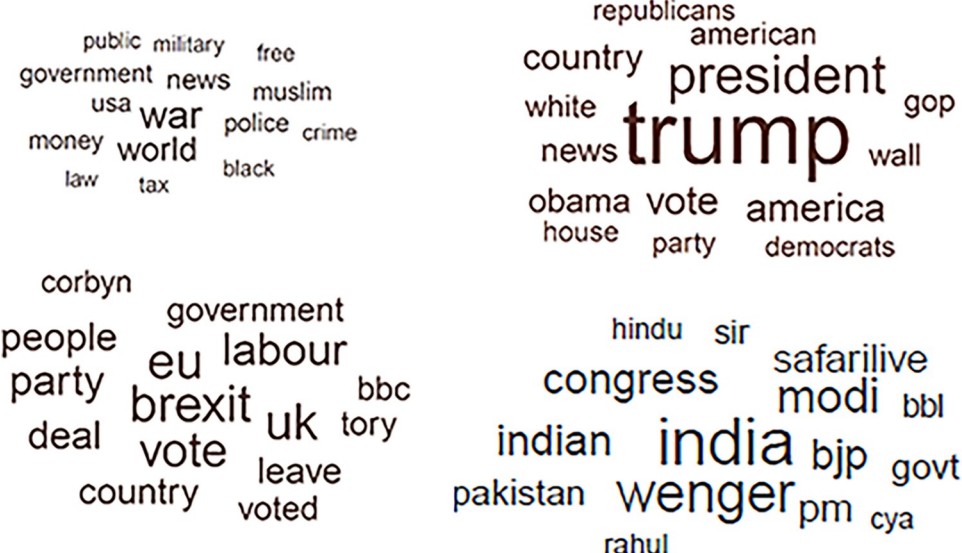

**Fig 4. Topics related to politics.** The larger the word, the higher the conditional probability of a word being in a topic.

general, with top words such as "war", "world", and "police" (topic 37). The second topic reflects US politics, with top words such as "trump", "president", "wall", "gop" (topic 40). The third topic reflects UK politics, with top words such as "brexit", "eu", "labour" (topic 31). The last topics is related to Indian politics, with top terms such as "india", "modi","bjp", "congress" (the last two terms refer to the mainstream Indian political parties; topic 12). Importantly, all of these topics were positively correlated with age, i.e., it seems that older people are more likely to discuss politics online, or on Twitter specifically.

Next, we aimed at selecting words from the four political topics with non-negligible conditional probabilities, which we set at 0.0001. Although this probability seems small, it is important to mention that the probabilities per each topic contains probabilities for nearly half a million terms. Thus, the per-word-per-topic probabilities are small, for example, the highest per-word-per-topic probability for the political topics was .03 for the term "trump". Out of the selected political words, 329 overlapped with the positive and 372 with the negative NRC wordlists.

However, excluding all of these words would not be justified as maybe only a small subset is responsible for inconsistency between LIWC and NRC scores among older adults. Instead, we aimed at identifying and excluding only those NRC words that do not correlate with LIWC affect scores in our Twitter sample. Thus, we correlated the frequency of each word with the respective LIWC scores: if a word is negatively correlated with the respective LIWC scores, it is likely that it causes inconsistency between LIWC and NRC scores.

For the NRC negative affect words, we found only 22 words negatively correlated with LIWC negative affect scores, e.g., "parade", "infamous", "wait". However, 187 NRC positive affect words were negatively correlated with LIWC positive affect scores. Among them there were words related to law, e.g., "attorney", "legal", "investigate"; the economy, e.g., "money", "pay", "wages"; politicians: e.g., "president", "cabinet", "mayor"; elections: "vote", "candidate", "ballot"; army: "veteran", "comrade", "intervention". Most interestingly, however, there were also several words representing values, such as: "democracy", "justice", "equality", "independence", "ethics", "conscience", "decency", which were negatively correlated with LIWC scores. Similarly, we found negative correlations with LIWC scores for seemingly positive words

related to benevolence, such as "innocent", "aid", "child", or "protect". The full list of the negatively correlated words is available in the S4 Table of S1 File. The NRC positive affect words were not only more often negatively correlated but these correlations also seem stronger than the correlations for the NRC negative affect words. As a next step, we excluded the selected terms from the NRC lexicon and re-computed NRC scores for overall, positive, and negative affect.

The re-analysis revealed that, again, the models with linear and quadratic effects of age outperformed models with linear age the only effect for overall, positive, and negative affect (the details are presented in S1 Table of S1 File). The predicted scores for all three models are shown in Fig 1 (third row). Removing identified negatively correlating political words substantially reduced the positive affect scores of people aged over 50. Now the predictions for positive affect by age are similar between NRC and LIWC. Unsurprisingly, the negative affect scores were less influenced by excluding political words: according to the NRC predictions, older people express the most negative affect, whereas LIWC predictions suggested young people express the most negative affect. Nevertheless, both methods predict an increase in negative affect from middle age onwards. Predicted scores for the overall affect are also similar across NRC and LIWC.

## Discussion

Quantitative text analysis has become widely adopted in social sciences and computer science, which is evident in recent publications in top journals [22, 29, 78–84]. One of the most popular themes of automated text analysis focuses around patterns of emotional expression [22, 29, 85–89]. As already mentioned, researchers can choose from multiple independently developed lexicons measuring emotional expressions. Although different lexicons usually correlate positively with one another, they do not entirely correlate [7] and it is not clear if each lexicon's correlations with external variables are comparable. It is possible that wordlists in specific contexts introduce biases in the relationships with the third variable. In this paper, we demonstrated such a scenario under which a researcher who made a seemingly inconsequential decision to use either the LIWC or the NRC lexicon to verify their question on life-span trajectories of emotional expressions would inadvertently be led to different theoretical conclusions depending upon the lexicon chosen.

### Emotional expression across age

In the analysis of the overall affect (negative scores subtracted from positive), both methods showed relatively consistent results. Specifically, we found that age affects emotional expressions in a quadratic manner: the relationship between age and emotional expressions was an inverted U-shape with a peak in the mid-50s. However, the NRC lexicon predicted that young people have the lowest overall affect scores, whereas the LIWC lexicon predicted that the youngest and oldest people had similarly low scores. But when we separately looked into positive and negative affect, which was critical for the theory testing, the LIWC and NRC lexicons differed to a rather large extent.

We observed substantially different patterns for positive affect for people above 50 years old depending on which lexicon we used. As the goal was to test the ageing theory of emotional experience, these findings are rather problematic. On one hand, the NRC scores are in line with previous studies showing that positive affect increases with age and reaches a plateau in older age or slightly decrease in very old age [45, 48]. In contrast, LIWC predictions are largely against the dominant theories. Specifically, we found an inversed U-shaped relationship between age and LIWC positive affect scores. In this case, a researcher would make different conclusions depending on whether they would have used the NRC or LIWC lexicon.

For the negative affect both LIWC and NRC lexicons showed a U-shaped relationship with age. Specifically, we observed a decrease in negative affect until middle age and further increase into older age. However, the two lexicons differed in the predictions of which age group would demonstrate the highest levels negative affect: LIWC predicted that the most negative are young people, partially supporting ageing theories such as Socio-emotional Selectivity Theory [8], but NRC suggests the highest negative affect should be expected for people above 65 years old. These findings pose difficulty for theory testing as, under the two scenarios, researchers would also come to different conclusions.

## Understanding the inconsistencies

To understand why we obtained different results from the LIWC and NRC lexicons, we investigated which words contributed to the LIWC and NRC scores. First, we found that the NRC Emotion Lexicon contains around 80 words which are classified as both positive and negative. This is not a large number given the fact that NRC has more than five and a half thousand words, i.e., these cross-classified words constitute only around 1.5% of the lexicon. However, we exclude these words from analysis because whether the word is reflective of positive or negative affect depends on the context of the sentence in which it is used, and this goes against the simple word counting approach of lexicons. The problem could arise if, for example, one group of study participants uses cross-classified words extensively and other groups use them scarcely. In this case, the overall affect score would not be affected (the effects would cancel each other out) but the positive and negative scores would be inflated for one group. For example, in our study, older adults used the word "vote" much more extensively than young people so both their NRC positive and negative affect scores might have been inflated.

Furthermore, our analyses found that NRC includes a considerable number of words which are related to politics. Again, the political words would not be an issue, if there were no differences between the age groups in the frequency they use those words. However, that was not the case: older versus young people disproportionately used political words in their tweets. This finding seems to be in line with data showing that among US adult Twitter users, people over 50 produce nearly 80% of the political tweets [90]. Removing identified political words substantially decreased NRC positive affect scores for older people, making the model predictions from LIWC and NRC more consistent.

Importantly, some of the political words in the NRC positive affect wordlist we excluded represent or are associated with values, e.g., "democracy", equality", "justice", or benevolence, e.g., "protect", "aid", which might seem odd because on a face of it these words appear quite positive. However, although social media are not a domain with very specialized content or a vocabulary, it seems that the political discussions on Twitter might provide a particular context which could bias meaning of such words. Specifically, it seems that people's networks are highly ideologically polarized when it comes to tweeting about politics [91]. Furthermore, people are likely to use highly emotional and moralized language when discussing political topics on Twitter [92, 93]. Thus, it is possible that in the context of Twitter, words that represent values might predominantly be associated with criticizing the current political situation or used as part of moralized conflicts between people supporting different political ideologies. For example, tweets containing these words might reflect concerns with democracy, justice, or ethics, rather than expressing these values per se. The additional analysis providing initial support for these claims is available in S5 and S6 Tables of S1 File where we found that, for example, the word most frequently used with "credibility" is "lost" (i.e., as in "lost credibility") and some of the top 10 words most frequently used with "oath" are "lied"/"lying"/"investigation". Future work could shed more light on this topic.

In contrast, for the negative affect, we think that the more extensive wordlist of NRC allowed for capturing words which had the negative meaning but were not included in LIWC. These were, for example, "bigot", "expense", or clearly political in nature, e.g., "government", "impeachment", "communism". Overall, the political words classified as negative in NRC, we suggest, helped in quantifying negative emotional expressions among people in older age. Importantly, these findings speak to the issue of the presumed context-independence of human-labelled lexicons.

Context-independence makes a lexicon applicable to many different contexts, but on the other hand, it might also pose a threat to research conclusions as we found in our analysis. Furthermore, the authors of NRC noted that "annotations for negative polarity have markedly higher agreement than annotations for positive polarity. This too may be because of the somewhat more fuzzy boundary between positive and neutral, than between negative and neutral" suggesting that users of NRC should be especially cautious when using its positive wordlists. Thus, sometimes might be beneficial to use lexicons that provide continues scores of how strong a word is associated with an emotion (e.g., [4, 94] rather than all-or-nothing method involved in word-counting approach.

In this paper we showed that employing more than one method could alleviate the potential shortcomings of just one lexicon and learn more about the data underlying the emotion lexicon scores. Furthermore, although the present research is not intended to be a tutorial on how to execute an analysis of the differences between two (or more) lexicons, we believe we presented an illustrative example of how to approach such a problem. We recommend that researchers examine carefully the words that contribute to the emotion scores produced using lexicons, e.g., by looking at top terms by a group they are interested in such as across age, gender, political ideology, etc. This is line with recommendations that when using lexicons, it is crucial to understand which words contribute to the overall lexicon scores [5, 86] and "remove entries from the lexicon that are not suitable (due to mismatch of sense, inappropriate human bias, etc.)" (p.3) [95]. As a next step, we think it is useful to understand what *classes of words* might be contributing to the differences if more than one lexicon is available. Here we ran LDA analysis to discover such classes. Later, researchers need to make an informed decision on if they want to exclude words and what is the basis for such an exclusion. Here, we correlated identified political words with LIWC scores because our goal was to understand the inconsistency between the two lexicons.

It is worth adding that a recent paper compares different lexicons and open methods of text analysis including their correlation with gender, age, and ideology [7]. In contrast, we focus on comparing just two lexicons in a greater details in response to a research question and illustrate how seemingly unimportant decision might lead to different conclusions, important for theory verification. We also show how one could approach such a problem. We agree with the recommendation Eichstaedt and colleagues [7] suggest: researchers should routinely use more than one method when analyzing textual data.

## Limitations

Before presenting final conclusions, we would like to make several observations on our findings. First, we present a one illustrative example of theory testing using two emotion lexicons and based on one dataset (which might be prone to a selection bias due to a method with which we harvested the data). Hence, the question of how common the problem might be and to what extent it applies to other research questions, methods, or datasets seems warranted. At this point we do not know the answer yet and believe future research could help shedding more light on this topic, e.g., by focusing on whether the classes of lexicons (e.g., denotative

lexicons vs. word-emotion associations) provide similar results when it comes to samples representing different demographics (e.g., age, gender, education, ideology), from different platforms (e.g., Twitter vs. Facebook), or on different topics (e.g., specialized topics vs. general). This, however, is beyond a scope of a current paper. Instead, we suggest that researchers should go beyond using just one lexicon and generally be reflective on the measures they use.

Relatedly, we would like to highlight that the current study is not a critique of any of the lexicons: we foresee that similar issues might be expected whichever lexicons we had chosen to use. Instead of concluding that one method is superior over another, we would like to caution the researcher to carefully examine the words that contribute to their emotion scores. Importantly, lexicons might also age and the usage of words might change over time. That was the case for the word "trump" which was part of the positive affect in NRC wordlists. Thus, it is another reason to inspect what words contribute to the emotion scores.

Finally, we should note that analysis of emotional expression from text entails some important ethical considerations. Some of these considerations apply to any method of analysis of emotional expression using natural language processing, e.g., danger of perpetuating biases against certain groups [96] or socio-cultural biases when methods are used outside the cultural context where they were created in, or the appropriate interpretation of scores (e.g., expression of sadness vs. being sad) [95]. Other considerations apply mostly to lexicons such as that word meanings might change depending on the context and time, or already mentioned differences between word-emotion association lexicons vs. denotative lexicons. These and other ethical considerations should be taken into account when planning and executing textual analysis [97].

## Conclusions

In this research we focused on the potential consequences of a common decision that researchers face while analysing emotional expression using lexicons: which lexicon to choose from the many available. In the example presented above, we have shown that two lexicons, LIWC and NRC, might disagree in their affective scores and that the small differences lead to different conclusions about the theory we tested.

We conclude that using a single lexicon might lead to unreliable conclusions, so we suggest that researchers should routinely use at least two lexicons created using different methodologies (e.g., expert-labelled, crowd-sourced, vs. machine-learned lexicons; or denotative vs. connotative lexicons). If both lexicons come to the same conclusion then the research evidence is reliable, but if not then researchers need to examine the lexicons to find out what difference might be causing the inconclusive result. If only one lexicon is available for a given trait, still we suggest that it is crucial to understand which words are highly contributing to the resulting scores. This, we believe, will lead to stronger conclusions from studies that involve lexicons.

## Supporting information

**S1 File.**
(DOCX)

## Acknowledgments

We would like to thank Sylwia Adamus, Rachela Antosz-Rekucka, Monika Biały, Jakub Cacek, Zuzanna Garncarek, Katarzyna Hryniewiecka, and Ewa Ilczuk from Jagiellonian University who contributed to data collection.

## Author Contributions

**Conceptualization:** Gabriela Czarnek.

**Data curation:** Gabriela Czarnek.

**Formal analysis:** Gabriela Czarnek.

**Funding acquisition:** Gabriela Czarnek.

**Investigation:** Gabriela Czarnek, David Stillwell.

**Methodology:** Gabriela Czarnek, David Stillwell.

**Supervision:** David Stillwell.

**Visualization:** Gabriela Czarnek.

**Writing – original draft:** Gabriela Czarnek.

**Writing – review & editing:** Gabriela Czarnek, David Stillwell.

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
