## [Decision Letter · Decision Letter 0]

30 Jun 2022

PONE-D-22-09013Two is better than one: Using a single emotion lexicon can lead to unreliable conclusionsPLOS ONE

Dear Dr. Stillwell,

Thank you for submitting your manuscript to PLOS ONE. After careful consideration, we feel that it has merit but does not fully meet PLOS ONE’s publication criteria as it currently stands. Therefore, we invite you to submit a revised version of the manuscript that addresses the points raised during the review process.

 We have received the reports from our advisors on your manuscript and read them carefully. We think the reviewers provided very good assessments and recommendations. Regarding the content of the reviews, we find no inconsistencies and we share their comments. Based on the advice received, we feel that your manuscript could be reconsidered for publication should you be prepared to incorporate revisions.

You should appreciate that this is a *major* revision. Minor changes to your manuscript will not be acceptable. Furthermore, the current decision does not guarantee an eventual acceptance of your paper - thus, the importance of your revisions.

We look forward to receiving your revised manuscript.

Kind regards,

Andrea Fronzetti Colladon, Ph.D.

Academic Editor

PLOS ONE

Journal Requirements:

2. In your Methods section, please include additional information about your dataset and ensure that you have included a statement specifying whether the collection and analysis method complied with the terms and conditions for the source of the data.

Reviewers' comments:

Reviewer's Responses to Questions

**Comments to the Author**

1. Is the manuscript technically sound, and do the data support the conclusions?

Reviewer #1: Yes

Reviewer #2: Yes

2. Has the statistical analysis been performed appropriately and rigorously? 

Reviewer #1: I Don't Know

Reviewer #2: Yes

3. Have the authors made all data underlying the findings in their manuscript fully available?

Reviewer #1: Yes

Reviewer #2: Yes

4. Is the manuscript presented in an intelligible fashion and written in standard English?

Reviewer #1: No

Reviewer #2: Yes

5. Review Comments to the Author

Reviewer #1: Review to: Two is better than one: Using a single emotion lexicon can lead to unreliable conclusions.

This is an interesting paper challenging the habit of just using a proven lexicon. The paper shows that when using different lexicons that where developed for a similar goal, there might be differences in outcome and they show in a small example on investigating where such differences in outcome come from: in their case: political words.

However, I think that the paper is a bit unclear on it’s message and what it’s main goal is.

In general the paper is quite vague which makes it difficult to understand. It is hard to point exactly to why, but I will point at two points where I think it needs improvement.

The two main points of vagueness: The influence of Political words, Conclusions

The influence of Political Words.

In itself this is an interesting section, and it shows the use of a large selection of methods. However, it’s function in the paper is not so clear. It is also not mentioned in the goals or introduction or abstract. The best description of what this chapter tries to achieve lies in the last sentence of the article: “but if not, then researchers need to examine the lexicons to find out what difference might be causing the inconclusive result.

This section is very dense in information compared to the other parts of the paper. A lot of methods pass by (LDA, LASSO, autoregressive multilevel model), but it is not clear why a certain method was employed and what the results of the method were and that the results are only . This makes this section hard to comprehend.

I would suggest to rewrite this section with more focus to what the method tries to achieve and less attention to details. For example, take more time to explain what the LDA does and why you are using it and less on details such as the amount of iterations (readers can find that in the code if this is published.) or the fact that you used Gibbs sampling (as far as I know this is the most often used). Only mention these details if you diverted from the default for a very specific reason (e.g. if Variational Bayes did not work in your occasion)

I would also suggest integrating this part better by more explicitly referring to it in the introduction/goals.

Conclusions:

The authors spend a significant part of the conclusions explaining what the paper is not. It is not a tutorial and it is not a critique on the lexicons. I would suggest to rephrase the conclusion such that it tells more about what the authors think the paper achieves instead of what they think that it doesn’t.

Smaller points/suggestions

- Although it is not necessary, It might be interesting to include a small discussion on the difference between Emotion analysis and Sentiment analysis.

- Figure 1: I would suggest to replace the titles Panel A, Panel B, Panel C with more descriptive labels: e.g. LIWC, NRC and NRC no political words.

- Figure 1: Produce the image in a better resolution. Currently it is not possible to read the axis ticks, even if you zoom in. Alternatively increase the font size of the ticks (and labels).

- line 219: how many tweets were deleted in each step?

- Line 241: Include a histogram to depict the distribution of ages.

Conclusion

Substantively I see the value of the paper. It compares and evaluates two lexicons that are often used without questioning. But I think the paper can be written more clearly such that it is clear what the authors did and why, as well as to properly evaluate the quality of the work.

For this reason I think a revise and resubmit would be the most appropriate way forward.

Reviewer #2: This is an interesting study that compares the use of two separate widely used emotion lexicons in a particular research project. It describes both how some conclusions are in alignment as well as how some conclusions are different when using the two lexicons for the project. Further, the paper examines, why some of the conclusions are different. This is a useful paper to highlight the nuances of emotion lexicon use in a research project.

There are three main points of improvement that I would suggest:

1. Some of the claims made towards the end of the paper about why the differences exist are not supported by evidence. For example, it is claimed thatwords like democracy, empathy, and equality are positive but that they are in reality used in negative tweets. Firstly, this needs to be supported with more evidence, than a mere citation to a paper. Secondly, if these words are primarily used in negative tweets then all that negativiness from the rest of those tweets will overshadow the positiveness in the overall affect curve. So in the end it just seems like, what is done here is to remove a large number of positive words that older people use, and that has reduced the positivity score when using NRC to bring the curve closer to LIWC.

Another claim without evidence: "several terms without clear emotional meaning are related to politics, e.g., for the positive NRC wordlist: “president”,“police”, “cabinet”; for the negative NRC wordlist: “government”, “foreign”, “liberal”"There is no clear fixed boundary between coarse categories like neutral and positive. In the NRC Lexicon, some majority of human annotators have said that "president" is associated with the positive class. For this paper to now claim/imply that the NRC lexicon should have only marked a word as positive if it has "clear emotional meaning" is a strawman argument. State clearly, what the NRC lexicon captures. Then if one wants to say that for an application that should only make use of terms that denotate an emotion, they shoudl not use the NRC lexicon, that is fair.

2. Lack of clear description of how the two lexicons were created. There is no description of how the NRc lexicon captures word--emotion associations (connotations); and how such lexicons are different from denotative lexicons. There is vague hand-waving that the LIWC is "theory-drive" and created by experts, but pretty much no description of the criteron for inclusion of a word in a category. No descriptions of whether these words are meant to denotate an emotion or can include words that are simply strongly associated with the emotion. The nature of a lexicon greatly impacts what conclusions one can draw from experiments that use the lexicon. This is of much greater significance than portrayed in the paper. The difference in results is not just a matter of coverage in the two lexicons.

3. Attributing gender by appearance can lead to misgendering. Even though te work here involves aggregate-level analysis, the paper should not perpetuate hetero-normative ideas of assuming gender by appearance or name.

4. There is no discussion of ethical considerations involved when suing emotion lexicons to draw inferences about people. See:Practical and Ethical Considerations in the Effective use of Emotion and Sentiment Lexicons.

Detailed comments:

"Thus, in this study,

74 we examine what would be the conclusions of a study testing a single research question using two

75 different and widely used emotion lexicons. Would the two methods give the same answer?"

How generic is this result based on one study?

The abstract and intro talk about:       theory driven lexicons and data-driven/crowdsourced lexicons at some points and,       theory driven lexicons and machine learned lexicons at other places. The machine learned lexicons come a bit out of the blue and need to be introduced appropriately.

How exactly does one create a "Theory-diven" lexicon. "Chosen by experts based on theory" seems somewhat vague.

In the context of small-LIWC-like lexicons and large NRC-Emotion-like lexicons, it is useful to talk about denotative and connotative lexicons. See Ref.Does LIWC mostly include words that denote emotion?NRC Emotion Lexicon includes word--emotion associations (connotations).Connotative expressions subsume denotative expressions and are much more common than denotative expressions. This seems like a key reason behind the success of connotative lexicons to analyze emotions in text.

"against a criterion such as emotion ratings (4–7)"emotion ratings of what?

"The 2015 version of LIWC has 6,400 terms within 90 pre-defined categories,"

cite corresponding paper. Point to where one can access the lexicon. Is the lexicon free for use?

"There is also a newer version of LIWC which has been released in February 2022."How is the 2022 version different from the 2015 version?

"psychological scales such as the PANAS (23)"How were words included in psychological scales such as the PANAS? Giving some idea of which words were selected and how representative they are of emotion words in general will be helpful.

How is "“goodness of fit” to a particular category defined?What is the goodness of fit chosen to be? e.g., the word should denote that category? Be strongly associated with that category? something else?

"the NRC emotional wordlist is about 4 times longer than LIWC’s" Is this including all LIWC words like past tense words or only words pertaining to emotions?

"The NRC Emotion Lexicon provides a list of 6,388 unique words (13,901 words in total,"Cite the paper for the lexicon. And point to the creators' website that distributes the lexicon:http://saifmohammad.com/WebPages/NRC-Emotion-Lexicon.htm

If I recall correctly, the lexicon has 14,000+ unique words marked as associated or not associated with various emotion categories.The R package is a third party product and not a proper reference when describing the lexicon. The R package probably is only using the words that are marked as associated with an emotion category (these might be ~6000).

"difference between the LIWC and NRC lexicons"One key difference is also that the NRC lexicon tries to capture how people think or use a word by asking a large number of people (and in that sense has a descriptive nature of language), whereas expert annotations tend to be about how a word should be used. Although, it is unclear how the LIWC annotators made their decisions.

"users who publicly disclosed their age"Worth noting that this might lead to a selection bias -- people on Twitter that are likely to disclose their age in a tweet.

"gender of the users was noted240 by the research assistants manually inspecting the accounts for clues such as their profile picture or

241 name)."Attributing gender by appearance can lead to misgendering. Even though te work here involves aggregate-level analysis, the paper should not perpetuate hetero-normative ideas of assuming gender by appearance or name.

Fig 1: Use the same scale on the Y axis (just as the same scale is used for x axis) for all the sub-figures. Otherwise, somebody who does not look at the y axis labels (which are out of focus and hard to read, BTW) can be easily misled.

When using large association lexicons, removing words not appropropriate is recommended:cite paper

Key Related work:PoKi: A Large Dataset of Poems by Children

"However, excluding all of these words would not be justified as they may accurately convey

404 positive or negative affect."It is not clear that removing "words associated with politics" is appropriate either.

"Instead, we aimed at identifying and excluding only the words that do not

405 correlate with affect in our Twitter sample."What does this mean?

Thus, we correlated the frequency of each word with the

406 respective LIWC scores: If a word is negatively correlated with the respective LIWC scores, it is likely407 that in our sample it does not convey the emotion suggested by the NRC lexicon."

This just seems like you are removing entries from the NRC lexicon that are not aligned with LIWC! This paper is trying to compare NRC Lex and LIWC. In such a case, LIWC cannot be taken as the "right" lexicon.

"The majority of these words are rather neutral when out of context."?How do you determine the boundary between neutral and positive? Is this simply a matter of what the authors of this paper feel? is the argument that the authors' boundary is the right one compared to the one determined by the annotators? Does the boundary change based on the application at hand?

I would argue these boundaries are artificial, and it is better to use relative sentiment lexicons such as the NRC VAD lexicon or the NRC Emotion Intensity lexicon. Ther the only claim is that one word is more positive than another. The claim is not that a word is "positive" or "neutral".

"Importantly, some of the political words in the NRC positive affect wordlist we excluded represent488 values, e.g., “democracy”, equality”, and “empathy”. Although those words have a positive

489 connotation in general, in the context of Twitter they are likely to be associated with criticizing the

490 current political situation or used as part of moralized conflicts between people supporting different

491 political ideologies (e.g., Sterling & Jost, 2018)."

what exactly do Sterling & Jost, 2018 show to support the claim made in this paper.

6. PLOS authors have the option to publish the peer review history of their article (what does this mean?). If published, this will include your full peer review and any attached files.

Reviewer #1: No

Reviewer #2: No

---

## [Author Response · Author response to Decision Letter 0]

31 Aug 2022

Reviewer #1:

Review to: Two is better than one: Using a single emotion lexicon can lead to unreliable conclusions.

This is an interesting paper challenging the habit of just using a proven lexicon. The paper shows that when using different lexicons that where developed for a similar goal, there might be differences in outcome and they show in a small example on investigating where such differences in outcome come from: in their case: political words.

However, I think that the paper is a bit unclear on it’s message and what it’s main goal is.

Thank you for this comment. In order to increase the clarity of the research goals, we streamlined the abstract to remove unnecessary information (e.g., sources of online text data) and instead provided more details on the method and results.

We have also shortened and streamlined the “Goals” section in the Introduction: we removed non-crucial information (e.g., general information about what language conveys and what can be studied using text data) and moved detailed regarding our methods to other sections (the definition and description of the emotion lexicons was moved to the next section; comparison of LIWC and NRC was moved to the section focusing on these lexicons). We believe now, the goals of this research are clearer. 

In general the paper is quite vague which makes it difficult to understand. It is hard to point exactly to why, but I will point at two points where I think it needs improvement.

The two main points of vagueness: The influence of Political words, Conclusions

1. The influence of Political Words.

In itself this is an interesting section, and it shows the use of a large selection of methods. However, it’s function in the paper is not so clear. It is also not mentioned in the goals or introduction or abstract. The best description of what this chapter tries to achieve lies in the last sentence of the article: “but if not, then researchers need to examine the lexicons to find out what difference might be causing the inconclusive result.

In order to improve the clarity, we added information on why we aimed to identify political words in the first paragraph of that section:

(p.18-19): “Given words’ contribution to the emotion scores presented above, it seems that the political words in the NRC could be responsible for the differences between the NRC and LIWC scores among older people. In order to better understand whether this is the case, we aimed at excluding political words which might contribute to the differences between NRC and LIWC model predictions.”

1.1

This section is very dense in information compared to the other parts of the paper. A lot of methods pass by (LDA, LASSO, autoregressive multilevel model), but it is not clear why a certain method was employed and what the results of the method were and that the results are only . This makes this section hard to comprehend.

I would suggest to rewrite this section with more focus to what the method tries to achieve and less attention to details. For example, take more time to explain what the LDA does and why you are using it and less on details such as the amount of iterations (readers can find that in the code if this is published.) or the fact that you used Gibbs sampling (as far as I know this is the most often used). Only mention these details if you diverted from the default for a very specific reason (e.g. if Variational Bayes did not work in your occasion)

I would also suggest integrating this part better by more explicitly referring to it in the introduction/goals.

We added a brief overview of the goals of the analysis presented in “The influence of political words” section. We believe this provides a non-technical intuition of what we were trying to achieve in this analysis:

(p.19) “The logic of this analysis is the following: we aimed to identify groups of words that are related to politics and frequently used by people in older age (because only those seem to be creating a difference between NRC and LIWC scores); then, we try to select the subgroup of these words that could be causing the inconsistency between NRC and LIWC by correlating the frequency of using these words with the LIWC emotion scores.” 

Furthermore, in line with Reviewer’s suggestions, we streamlined the description of the LDA analysis by removing unnecessary details. Now the rewritten paragraph says: 

(p. 19) “We first excluded stopwords, i.e., frequent but not informative words (e.g., we, the, be) [57] and 27,414,128 terms remained for the LDA analysis. After identifying the appropriate number of topics using several criteria [74–77] with the ldatuning package [78] (see the Supporting information S3 Fig for details), we modelled 50 topics with the topicmodels package [79].”

We also added less technical and more narrative description of each analytic step before presenting the analysis, e.g., 

(p. 19) “However, including all of the 50 topics in the subsequent analyses could lead to over-fitting, multicollinearity, and thus problems with a model interpretation. Hence, we decided to trim the number of topics using penalized regression (LASSO regression).”

(p. 19) “Because LASSO regression did not allow us to use all available data (we could not properly account for a multilevel structure of our dataset), we next used an analytic technique that allows for a more comprehensive analysis of what topics are more frequently used in older age.”

(p. 20) “Next, we aimed at selecting words from the four political topics with non-negligible conditional probabilities (…).”

2. Conclusions:

The authors spend a significant part of the conclusions explaining what the paper is not. It is not a tutorial and it is not a critique on the lexicons. I would suggest to rephrase the conclusion such that it tells more about what the authors think the paper achieves instead of what they think that it doesn’t.

 When presenting the conclusions, our goal was to make it clear that we do not want to criticize any lexicon. To the contrary, we believe researchers might find themselves in a similar conundrum when using two lexicons that lead them to different conclusions regarding their research questions. However, we agree that highlighting this issue at the end of the paper might dilute the main message. Thus, we moved the part on what this paper “does not do” earlier in the Discussion section, and shortened the Conclusion section to only two key paragraphs. 

Smaller points/suggestions

3.

- Although it is not necessary, It might be interesting to include a small discussion on the difference between Emotion analysis and Sentiment analysis.

 We added a clarification regarding what affect vs. specific emotions are as well as distinction between sentiment vs. emotion analysis. Specifically, we added the following sentences to the Introduction 

(p. 4): “It is worth noting that in psychology, the terms affect and emotions are not interchangeable: affect can be positive, negative, or neutral and experienced as free-floating (mood) or a short-lived state; whereas specific emotions, e.g., anger, contempt, are a complex pattern of physiological (e.g., hormones levels), cognitive (e.g., interpretations), and behavioural changes (e.g., facial expressions) in response to a particular object or situation [13,14]. In a similar vein, some researchers use the term “sentiment analysis”, which comes from early work on detecting subjectivity of customer reviews[15]; it refers to extracting polarity from texts (positive, negative, or neutral, similar to the notion of affect). Others differentiate it from “emotion analysis” which focuses on understanding specific emotions e.g., [16]. We use the terms “emotion lexicon” broadly, i.e., any list of words that can capture expression of feelings, affect, mood, sentiments, or specific emotion, similarly to [15].”

4.

- Figure 1: I would suggest to replace the titles Panel A, Panel B, Panel C with more descriptive labels: e.g. LIWC, NRC and NRC no political words.

- Figure 1: Produce the image in a better resolution. Currently it is not possible to read the axis ticks, even if you zoom in. Alternatively increase the font size of the ticks (and labels).

 We have now edited the Figure 1 in line with Reviewer’s suggestions: we have changed the labels of the panels and increased the resolution of the charts.

5.

- line 219: how many tweets were deleted in each step?

 We have added the detailed of how many accounts and tweets were deleted at each step of the analysis to the Data cleaning section. For example:

(pp. 12-13) “Starting with the user-level, we were able to retrieve age announcement tweets of 4,826 users out of 5,124 links harvested by research assistants (some tweets were removed, accounts became private, etc.). Next, we included only public Twitter accounts where users specified English as their language, which left 4,754 users in the dataset.” 

(p. 13): “On the tweet-level, we retained only original posts and excluded re-tweets (2.5 million out of nearly 8.5 million tweets).”

6.

- Line 241: Include a histogram to depict the distribution of ages.

 We have now added a histogram to the Supplementary materials in S1 Fig. In the main text, we added the following information:

(p. 13) “A histogram depicting distribution of age in the sample is shown in Supplementary materials S1 Fig.”

Conclusion

Substantively I see the value of the paper. It compares and evaluates two lexicons that are often used without questioning. But I think the paper can be written more clearly such that it is clear what the authors did and why, as well as to properly evaluate the quality of the work.

For this reason I think a revise and resubmit would be the most appropriate way forward.

Reviewer #2: 

This is an interesting study that compares the use of two separate widely used emotion lexicons in a particular research project. It describes both how some conclusions are in alignment as well as how some conclusions are different when using the two lexicons for the project. Further, the paper examines, why some of the conclusions are different. This is a useful paper to highlight the nuances of emotion lexicon use in a research project.

There are three main points of improvement that I would suggest:

1. Some of the claims made towards the end of the paper about why the differences exist are not supported by evidence. For example, it is claimed that words like democracy, empathy, and equality are positive but that they are in reality used in negative tweets. Firstly, this needs to be supported with more evidence, than a mere citation to a paper.Secondly, if these words are primarily used in negative tweets then all that negativiness from the rest of those tweets will overshadow the positiveness in the overall affect curve. So in the end it just seems like, what is done here is to remove a large number of positive words that older people use, and that has reduced the positivity score when using NRC to bring the curve closer to LIWC.

 Thank you for this comment. In order to provide more evidence for our claims regarding the use of positive NRC words related to values, we expand our work in two ways. First, we argue that although Twitter is not a highly specialized domain, the political conversation might be somewhat different than conversations on non-political topics. For example, the social networks seem largely ideologically polarized when it comes to political conversations in comparison to conversation on non-political topics. Furthermore, people are likely to use moral and emotional language when it comes to political discussion. This provides a specific context for political discussions in societies that are highly polarized. The discussion is also generally toned down as compared to the previous version. We now state:

(p. 24): “Furthermore, our analyses found that NRC includes a considerable number of words which are related to politics. The political words would not be an issue, if there were no differences between the age groups in the frequency they use those words. However, that was not the case: older versus young people disproportionately used political words in their tweets. This finding seems to be in line with data showing that among US adult Twitter users, people over 50 produce nearly 80% of the political tweets (93). Removing identified political words substantially decreased NRC positive affect scores for older people, making the model predictions from LIWC and NRC more consistent. Importantly, some of the political words in the NRC positive affect wordlist we excluded represent values, e.g., “democracy”, equality”, “justice”, or benevolence, e.g., “protect”, “aid”, which might seem odd because on the face of it these words might seem quite positive. However, although social media are not a domain with very specialized content or a vocabulary, it seems that the political discussions on Twitter might provide a particular context which could bias meaning of such words. Specifically, it seems that people’s networks are highly ideologically polarized when it comes to tweeting about politics (94). Furthermore, people are likely to use highly emotional and moralized language when it comes to political topics on Twitter (95,96). Thus, it is possible that in the context of Twitter, words that represent values might actually be associated with criticizing the current political situation or used as part of moralized conflicts between people supporting different political ideologies. For example, tweets containing these words might reflect concerns with democracy, justice, or ethics, rather than expressing these values per se.”

Secondly, to provide more evidence on the context in which the words representing values (or more broadly ethics, morality, or benevolence), we ran separate analyses that are presented in Supplementary materials Tables S6-S7. These analyses, inspired by work on collocation analysis, we select ten words which were related to values and had the highest negative correlation with LIWC, e.g., “justice”, “democracy”, or “oath”. This analysis is in line with our observations that value-related words might be used in negative context when part of political discussions on Twitter. We found that words such as “oath” are used in the context of words such as investigation and lying. Furthermore, because the value-related words might co-occur (e.g., “protect democracy”), the positive emotionality may not be superseded by the rest of the sentence. 

However, as suggested by Reviewer 1, the "The influence of the political words" section is already quite complex. Thus, we present the new analysis in the Supplementary materials. In the man text we also suggested that more work on the topic might be needed (e.g., analysis of words related to economy, law, elections, or institutions). In our newly added analyses, we focus on words related to values because they seem to provide a good test our hypothesis (strongly positive words used in a negative context). In the main text we mention the analyses. We have added to the main text:

(p. 24) “The additional analysis providing initial support for these claims is available in Supplementary materials Tables S6 and S7, where we found that, for example, the word most frequently used with “credibility” is “lost” (i.e., as in “lost credibility”) and some of the top 10 words most frequently used with “oath” are “lied”/”lying”/”investigation”. Future work could shed more light on this topic."

We would also like to highlight that our goal here is to understand inconsistency between the NRC and LIWC. We will elaborate on this issue in response to comment #22 below, where Reviewer provided specific concerns regarding our analysis. 

1.1

Another claim without evidence: "several terms without clear emotional meaning are related to politics, e.g., for the positive NRC wordlist: “president”,“police”, “cabinet”; for the negative NRC wordlist: “government”, “foreign”, “liberal”"There is no clear fixed boundary between coarse categories like neutral and positive. In the NRC Lexicon, some majority of human annotators have said that "president" is associated with the positive class. For this paper to now claim/imply that the NRC lexicon should have only marked a word as positive if it has "clear emotional meaning" is a strawman argument. State clearly, what the NRC lexicon captures. Then if one wants to say that for an application that should only make use of terms that denote an emotion, they shoudl not use the NRC lexicon, that is fair.

 We have now explained the denotation vs. connotation in the lexicons and described the differences between NRC and LIWC more clearly (the detailed changes are provided in response to comment #2).

Additionally, we have now changed the problematic sentences to avoid stating that some words do not have emotional meaning and instead focus on the inconsistency between NRC and LIWC scores. Specifically, the rewritten paragraph now says:

(p. 18) “However, in the NRC word clouds for older age several terms seem to be related to politics, both for the positive and negative affect, e.g., for the positive NRC wordlist: “president”, “police”, “cabinet”; for the negative NRC wordlist: “government”, “foreign”, “liberal”; (…) because the political words were not present in LIWC word clouds, they are a candidate class of words that could be responsible for the inconsistency in NRC and LIWC scores for people in older age. In the next section we will identify the political words that could be causing these inconsistencies.“

2. Lack of clear description of how the two lexicons were created. There is no description of how the NRc lexicon captures word--emotion associations (connotations); and how such lexicons are different from denotative lexicons. There is vague hand-waving that the LIWC is "theory-drive" and created by experts, but pretty much no description of the criteron for inclusion of a word in a category. No descriptions of whether these words are meant to denote an emotion or can include words that are simply strongly associated with the emotion. The nature of a lexicon greatly impacts what conclusions one can draw from experiments that use the lexicon. This is of much greater significance than portrayed in the paper. The difference in results is not just a matter of coverage in the two lexicons.

 Thank you for this comment. We now completely re-wrote the “Emotion lexicons” section in which we present a more comprehensive review of different methods of creating emotion lexicon. Specifically, we now clearly state that there are two methods of creating a lexicon: manual (human-labelled) vs. automatic (machine-learned) and give brief overview of these different methods. Furthermore, as per Reviewer suggestion, we added information on the distinction between denotative vs. connotative lexicon. Specifically, we add the following paragraphs:

(pp. 4-5): “Many lexicons are available, often measuring exactly the same emotions but differing by the number of terms they contain, the method of their creation, or the goals with which they were created. With regard to method of creation, lexicons could be divided into two broad categories: human-labelled vs. machine-learned. 

Human-labelled methods involve human annotators, experts or crowd-sourced, who provide an evaluation of emotional qualities of words which later become lexicon terms [17]. Experts are usually just a few psychology or linguistics professionals. Typically, they individually decide whether candidate words are an acceptable example of a certain emotional expression, and later discuss the inconsistencies between their ratings, and agree on a final lists of lexicon words, which is also supported by the psychometric analysis. In contrast, crowd-sourcing involves large number of lay people (e.g., MTurk workers, [11]) who provide ratings of several words (and not all the words); their rating are later aggregated and the statistical agreement determines the lexicon words. 

Additionally, some lexicons provide lists of words that denote emotions, that is words that express a particular emotion irrespective of the context, e.g., rage, great [15]. Other lexicons instead, are broader in the approach and focus on words’ emotional connotation, i.e., whether the words are associated with particular emotion, e.g., loss, friendship [15]. Importantly, words that denote emotions (e.g., rage) are also associated with the emotion, thus might be included in such a lexicon. The denotative lexicons have the advantage that the instances of false positives (e.g. a word incorrectly labelled as positive which is actually not) should be low, but connotative lexicons have more scope to identify a larger list of words so the instances of false negatives (e.g. a word incorrectly not labelled as positive which actually is) should be low. Nevertheless, because involving people, experts or non-experts, is rather costly and lengthy, these lexicons are usually not longer than a few hundred to thousand words [15]. In consequence, such lexicons might have low coverage of the total number of words in a text. Thus, automated are employed to creating word-association lexicons. 

Creating a lexicon with machine-learning typically involve a labelled dataset of sentences that express a certain emotion (training dataset); various techniques or their combination can be used to identify a list of words which correlate with the ground truth labelled data [17]. For example, researchers created a large dataset with tweets containing one of nearly 80 emotion-word hashtags; if a tweet had a hashtag denoting positive emotion (e.g., #great), it was considered being positive, and if it had a hashtag denoting a negative emotion (e.g., #terrible), it was considered being negative. Next, they used a machine-learning classifier on this training data (with a number of sematic, sentiment, and surface-form features) to create a lexicon containing entries for ~54 thousand unigrams (single words or terms) and ~317 thousand bigram (two-term sequences). Importantly, machine-learned lexicons provide word-emotion associations, i.e., they are connotative. Machine learning methods can be used to create lexicons that are suited to a particular domain given that words might have different emotional connotations given the context, e.g., [18] as opposed to more general human-labelled lexicons. For more comprehensive and non-technical overview of methods of automatic detection of emotions in texts, including emotion lexicons, see [15].”

Additionally, when introducing LIWC and NRC Emotion Lexicon in the further section (“Lexicons in the current analyses”), we refer to the distinction between denotative vs. connotative lexicons. Importantly, LIWC does not only provide words denoting particular emotions but it also contains words such as “accept”, “adventure”, or “kiss” for “positive emotion”, and “alone”, “neglect”, or “thief” for the “negative emotion” wordlist. However, in line with the Reviewer’s suggestion presented in the detailed comments below, it seems that the proportion of words expressing emotions vs. associated with emotions is larger in NRC Emotion Lexicon. Calculating such a proportion is not feasible due to the fact that LIWC provides words and word stems (so an exhaustive list of words would be difficult to provide to calculate such an exact proportion).

Additionally, while comparing the two lexicons, we also add the following sentence:

(p. 10): Importantly, although LIWC seem to be including both words expressing emotions (denotation), it includes also emotion-word associations that the judges were able to agree upon. In contrast, NRC focuses on word-emotion associations, i.e., it includes terms expressing particular emotions but the associations between a term and an emotion might be more or less close given that both “strong” and “moderate” associations were accepted.

3. Attributing gender by appearance can lead to misgendering. Even though the work here involves aggregate-level analysis, the paper should not perpetuate hetero-normative ideas of assuming gender by appearance or name.

We have now removed mentioning the gender throughout the paper and re-ran the analyzes without including gender; it affected the results presented in Figure 1 and S1 Table of the Supplementary materials. However, the changes are not substantial and all our conclusions regarding the data or the lexicons are still valid.

4. There is no discussion of ethical considerations involved when suing emotion lexicons to draw inferences about people. See:Practical and Ethical Considerations in the Effective use of Emotion and Sentiment Lexicons.

Thank you for mentioning this relevant paper. We were unaware of it at the time of writing this manuscript but now included in manuscript. We have also included a paragraph in the Discussion section mentioning the ethical consideration of emotion analysis from texts:

(p. 27) “Finally, we should note that analysis of emotional expression from text entails some important ethical considerations. Some of these considerations apply to any method of analysis of emotional expression using natural language processing, e.g., danger of perpetuating biases against certain groups [99] or socio-cultural biases when methods are used outside the cultural context where they were created in, or the appropriate interpretation of scores (e.g., expression of sadness vs. being sad) [98]. Other considerations apply mostly to lexicons such as that word meanings might change depending on the context and time, or already mentioned differences between word-emotion association lexicons vs. denotative lexicons. These and other ethical considerations should be taken into account when planning and executing textual analysis [100].”

Detailed comments:

5.

"Thus, in this study, we examine what would be the conclusions of a study testing a single research question using two different and widely used emotion lexicons. Would the two methods give the same answer?" 

How generic is this result based on one study?

 We added a comment on a limitations related to using just one research question, dataset, and two methods. We would like to highlight that the goal of this research was not to demonstrate how widespread the problem might be but rather signal it and suggest how we can make study conclusions robust. Specifically, we added the following paragraph to the Discussion section:

(p. 26) “(…) the question of how common the problem might be and to what extent it applies to other research questions, methods, or datasets seems warranted. At this point we do not know the answer yet and believe future research could help shedding more light on this topic, e.g., by focusing on whether the classes of lexicons (e.g., denotative lexicons vs. word-emotion associations) provide similar results when it comes to samples representing different demographics (e.g., age, gender, education, ideology), from different platforms (e.g., Twitter vs. Facebook), or on different topics (e.g., specialized topics vs. general). This, however, is beyond a scope of a current paper. Instead, we suggest that researchers should go beyond using just one lexicon and generally be reflective on the measures they use.”

6.

The abstract and intro talk about: theory driven lexicons and data-driven/crowdsourced lexicons at some points and, theory driven lexicons and machine learned lexicons at other places. The machine learned lexicons come a bit out of the blue and need to be introduced appropriately.

 We now removed discussion of theory vs. data vs. crowd-sourced lexicons from the Abstract as it does not provide enough space to introduce these methods properly. Instead, in line with Reviewer’s suggestion, we describe these methods in the “Emotion lexicons” section (pp. 4-6); which now integrates the information previously presented in the “Goals” and “Creating emotion lexicon” sections. The new section briefly introduces different methods and approaches to creating lexicons (human-labelled vs. machine-learned; denotative vs. connotative lexicons). The detailed information on the edits is provided in response to the comment #2 above.

7.

How exactly does one create a “Theory-diven” lexicon. “Chosen by experts based on theory” seems somewhat vague.

 As already mentioned, the revised version of the manuscript is now more precise when describing different methods of creating lexicons; we now refrain from using “theory-driven” approach altogether. The detailed information on the edits is provided in response to the comment #2 above.

8. 

In the context of small-LIWC-like lexicons and large NRC-Emotion-like lexicons, it is useful to talk about denotative and connotative lexicons. See Ref.Does LIWC mostly include words that denote emotion? NRC Emotion Lexicon includes word—emotion associations (connotations).Connotative expressions subsume denotative expressions and are much more common than denotative expressions. This seems like a key reason behind the success of connotative lexicons to analyze emotions in text.

 We have now added a discussion of differences between denotative vs. connotative lexicons. Furthermore, in line with the Reviewer suggestion, it seems that in LIWC, the proportion of words expressing emotions vs. associated with emotions is larger than in the NRC Emotion Lexicon. But as mentioned before (in response to comment #2), calculating such a proportion is not feasible due to the fact that LIWC provides words and word stems (so an exhaustive list of words would be difficult to provide to calculate such an exact proportion). The detailed information on the edits is provided in response to the comment #2 above.

9. 

"against a criterion such as emotion ratings (4–7)"emotion ratings of what?

 We clarified this sentences which now reads:

(p. 3): “To do this, typically researchers correlate the scores from different lexicons or correlate lexicon scores against a criterion such as human ratings of emotional expression in a particular text [4–7].”

10.

"The 2015 version of LIWC has 6,400 terms within 90 pre-defined categories," cite corresponding paper. Point to where one can access the lexicon. Is the lexicon free for use?

 We added the requested information. Specifically, we added: citation of the LIWC 2015 manual with a link to a website where LIWC can be purchased, information how to access the lexicon and what that there are fees associated with the lexicon use. The details are presented in our response to the next comment. 

11.

“There is also a newer version of LIWC which has been released in February 2022.”How is the 2022 version different from the 2015 version?

 We added a paragraph describing the practicalities of accessing and using LIWC as well as the differences between 2015 and newest 2022 version. Specifically, we added the following paragraph: 

(pp.7-8) “LIWC is available with an accompanying software for a moderate fee. Using LIWC does not require any programming skills so it is accessible to a broad range of users. The new, fifth, LIWC version has been released in February 2022 [19]; and for new users, it is only possible to purchase the fifth version through the LIWC app (through which 2015 and earlier LIWC versions can be accessed). The major differences between 2015 and the new 2022 LIWC version are: adding new categories (e.g., “ethnicity”, “fatigue”), expanding wordlists in the existing categories (it now has over 12,000 terms in total), and removing a few categories due to their low base rates (e.g., “comparison words”). For the emotion lexicons, the important changes include: replacing “positive emotion” and “negative emotion” categories with “positive sentiment” and “negative sentiment”, respectively (however, the respective lists are very similar and the correlations between 2015 and 2022 scores for a sample text is around 0.85 [19]); changing the content of specific emotion lexicons so that they now include only denotative words or strongly associated words; and excluding the swear words from positive and negative sentiment/emotion categories; the details are available in the 2022 LIWC psychometric manual [19]. The conclusions from the current paper which uses the 2015 LIWC version, we believe, are still important and illustrative of a larger issue related to the usage of emotion lexicons, which just does not pertain to a particular lexicon or its version; we elaborate on this issue in the Discussion section. Furthermore, given that LIWC 2022 has just been released, we expect many researchers still use the 2015 LIWC version.”

12.

"psychological scales such as the PANAS (23)"How were words included in psychological scales such as the PANAS? Giving some idea of which words were selected and how representative they are of emotion words in general will be helpful.

 With regard to specific examples taken from PANAS, they were not provided in the 2015 LIWC manual. Nevertheless, to make the manuscript more accessible to broad readership, we provided a few example of PANAS words that are included in LIWC lexicon. Additionally, we now expanded description of how LIWC was created. The details are provided in responses to the comment #13.

13.

How is "“goodness of fit” to a particular category defined? What is the goodness of fit chosen to be? e.g., the word should denote that category? Be strongly associated with that category? something else?

 Unfortunately, the Authors of the LIWC 2015 manual did not provide a definition or examples of what “goodness-of-fit” is; they only mention that the analysis was qualitative at this point. Given the words included in the lexicon, it seems that both denotative words and word-emotion associations were included. We have expanded the description of how LIWC was created to provide more comprehensive information to the readers. Specifically, the revised version states:

(p. 7) “Each category of LIWC, including the emotional lexicons, was created in several steps. During Word Collection, the emotional words were harvested from existing psychological scales (e.g., words such as upset or proud from PANAS [14]), their synonyms from a thesaurus were added; next a group of several judges generated new words individually and later in a group brainstorming sessions. Next, in the Judge Rating Phase, the collected wordlists were qualitatively evaluated by a panel of the judges. For a word to be included in a particular category, the majority of the experts needed to agree on its goodness-of-fit to that category; if they could not decide on the appropriate category, a word was removed from the lexicon. In the Base Rate Analysis, the frequencies of the words were evaluated using multiple sources of text data (e.g., blog posts, Twitter, spoken language); if the word did not occur at least once in several of those texts, it was removed from the lexicon. Next, in Candidate Word List Generation, new words were harvested from multiple texts from previous studies; if a word had high frequency, was not included in LIWC already, and was correlating with a LIWC category, several judges decided on suitability for inclusion of that word. In the Psychometric Evaluation, a psychometric analysis of each category was conducted; if a word was detrimental to a category’s internal consistency it was flagged, and the panel of judges again, decided on whether to keep it or remove it from the lexicon. All these steps were repeated in order to refine the lexicon and catch potential mistakes; however, the Authors note that the changes in this Refinement Phase were marginal. Overall, the process of creating the lexicon is largely based on the expert consensus but also seem time-consuming and rather high in costs.”

14.

"the NRC emotional wordlist is about 4 times longer than LIWC’s" Is this including all LIWC words like past tense words or only words pertaining to emotions?

 We clarified the length of both LIWC and NRC wordlists. 

Specifically, when describing the LIWC lexicon we added the following sentence: 

(pp. 6-7) “Importantly, in this paper we focus on positive and negative affect (and not on specific emotion); “positive emotion” and “negative emotion” lists have 1,364 terms in total. Some of the LIWC terms are stemmed words which expand the lexicon significantly (…).”

When describing NRC lexicons, we added (p. 9): “The “positive sentiment” and “negative sentiment” wordlists that we will focus on later, have 5,624 terms in total.”

Later, when comparing the two lexicons, we added (p. 10): “(…), for positive and negative affect LIWC provides around 1.3 thousand terms (including word stems) and NRC provides around 5.6 thousand terms.”

15.

"The NRC Emotion Lexicon provides a list of 6,388 unique words (13,901 words in total," Cite the paper for the lexicon. And point to the creators' website that distributes the lexicon: http://saifmohammad.com/WebPages/NRC-Emotion-Lexicon.htm

 We added the citation of the original paper as well as the link to the website. Specifically, we added:

(p. 8): “In contrast to LIWC, The NRC Emotion Lexicon [11,12] focuses on emotional words only, and provides a list of 13,872 terms in several categories.”

(p. 9): “NRC is freely available for academic use at [31] (…)”

Relevant excerpts from the References:

11. Mohammad SM, Turney PD. Emotions Evoked by Common Words and Phrases: Using Mechanical Turk to Create an Emotion Lexicon. In: Proceedings of the NAACL-HLT 2010 Workshop on Computational Approaches to Analysis and Generation of Emotion in Text. Los Angeles, CA; 2010. p. 1–9. 

12. Mohammad SM, Turney PD. Crowdsourcing a Word–Emotion Association Lexicon. Comput Intell. 2013;29(3):436–65. 

31. NRC Emotion Lexicon [Internet]. [cited 2022 Jul 10]. Available from: http://saifmohammad.com/WebPages/NRC-Emotion-Lexicon.htm

16. 

If I recall correctly, the lexicon has 14,000+ unique words marked as associated or not associated with various emotion categories. The R package is a third party product and not a proper reference when describing the lexicon. The R package probably is only using the words that are marked as associated with an emotion category (these might be ~6000).

 We clarified the number of entries in the NRC and as we were interested only in the analysis of positive and negative affect, we added the exact numbers of these terms. The details are provided in response to comment #14.

17.

“difference between the LIWC and NRC lexicons” One key difference is also that the NRC lexicon tries to capture how people think or use a word by asking a large number of people (and in that sense has a descriptive nature of language), whereas expert annotations tend to be about how a word should be used. Although, it is unclear how the LIWC annotators made their decisions.

 We agree that it is a bit unclear what exactly was the instruction to the judges employed to curate LIWC lexicon. Nevertheless, based on the available information, we provided overview of the methodology behind LIWC and NRC. The details are provided in response to comments #2, #6 - #8.

18.

“users who publicly disclosed their age”Worth noting that this might lead to a selection bias – people on Twitter that are likely to disclose their age in a tweet.

 We have mentioned that the dataset might be prone to a selection bias in the Discussion section:

(p. 26): “(…) we present a one illustrative example of theory testing using two emotion lexicons and based on dataset (which might be prone to a selection bias due to a method with which we harvested the data).”

19.

“gender of the users was noted240 by the research assistants manually inspecting the accounts for clues such as their profile picture or 241 name).”Attributing gender by appearance can lead to misgendering. Even though te work here involves aggregate-level analysis, the paper should not perpetuate hetero-normative ideas of assuming gender by appearance or name.

 We have now removed mentioning the gender or including it in the analyses. The details of the changes are presented in response to comment #3.

20.

Fig 1: Use the same scale on the Y axis (just as the same scale is used for x axis) for all the sub-figures. Otherwise, somebody who does not look at the y axis labels (which are out of focus and hard to read, BTW) can be easily misled.

 We have now updated the Figure 1 so it has better resolution and clearer panel captions (in line with Reviewer 1 comments). We, however, decided to keep the current axes as different models have different intercepts (due to different number of words matching the lexicons, lexicons length, etc.). Furthermore, this analysis does not remove stopwords, thus the total number of words is large and current formatting allows for easier spotting the differences between LIWC and NRC Emotion Lexicon predictions. 

21.

When using large association lexicons, removing words not appropropriate is recommended:cite paper.

Key Related work:PoKi: A Large Dataset of Poems by Children

 We have now added a suggested information with appropriate citation. Specifically, we have added: 

(p. 25) “This is line with recommendations that when using lexicons, it is crucial to understand which words contribute to the overall lexicon scores [5,89] and ‘remove entries from the lexicon that are not suitable (due to mismatch of sense, inappropriate human bias, etc.)’ (p.3) [98]”

22.

“However, excluding all of these words would not be justified as they may accurately convey positive or negative affect.”It is not clear that removing “words associated with politics” is appropriate either.

“Instead, we aimed at identifying and excluding only the words that do not correlate with affect in our Twitter sample.”What does this mean?

Thus, we correlated the frequency of each word with the respective LIWC scores: If a word is negatively correlated with the respective LIWC scores, it is likely that in our sample it does not convey the emotion suggested by the NRC lexicon.”

This just seems like you are removing entries from the NRC lexicon that are not aligned with LIWC! This paper is trying to compare NRC Lex and LIWC. In such a case, LIWC cannot be taken as the “right” lexicon.

 We rephrased the paragraph suggesting that some words are “right/correct” or “wrong” in the NRC Emotion Lexicon. Instead, we focus on identifying words that cause (or contribute to) inconsistency between NRC and LIWC scores among older adults. Specifically, the re-written paragraph now says: 

(pp. 20-21) “However, excluding all of these words would not be justified as maybe only a small subset is responsible for inconsistency between LIWC and NRC scores among older adults. Instead, we aimed at identifying and excluding only those NRC words that do not correlate with LIWC affect scores in our Twitter sample. Thus, we correlated the frequency of each word with the respective LIWC scores: if a word is negatively correlated with the respective LIWC scores, it is likely that it causes inconsistency between LIWC and NRC scores.”

Subsequently, we have also re-phrased the Results and Discussion sections that were using similar wording to focus more on consistency between LIWC and NRC rather on “accuracy”.

23.

"The majority of these words are rather neutral when out of context."?How do you determine the boundary between neutral and positive? Is this simply a matter of what the authors of this paper feel? is the argument that the authors' boundary is the right one compared to the one determined by the annotators? Does the boundary change based on the application at hand?

I would argue these boundaries are artificial, and it is better to use relative sentiment lexicons such as the NRC VAD lexicon or the NRC Emotion Intensity lexicon. Ther the only claim is that one word is more positive than another. The claim is not that a word is "positive" or "neutral".

 In the revised manuscript, we have removed the sentence suggesting that some of the NRC terms are neutral. 

Furthermore, in line with Reviewer suggestion, we have added information about the relative sentiment lexicons: 

(p. 25) “(…) sometimes might be beneficial to use lexicons that provide continues scores of how strong a word is associated with an emotion (e.g., (4,98) rather than all-or-nothing method involved in word-counting approach.”

24.

"Importantly, some of the political words in the NRC positive affect wordlist we excluded represent values, e.g., “democracy”, equality”, and “empathy”. Although those words have a positive connotation in general, in the context of Twitter they are likely to be associated with criticizing the current political situation or used as part of moralized conflicts between people supporting different political ideologies (e.g., Sterling & Jost, 2018)."

what exactly do Sterling & Jost, 2018 show to support the claim made in this paper.

 We have cited Sterling and Jost (2018) in the context of their analysis showing that when it comes to political discussions, the same words can be used with qualitatively different meaning. We have now expanded on this analysis in the Supplementary materials (more on this issue is provided in response to comment #1). In the Supplementary materials we have also elaborate further on Sterling and Jost (2018). Specifically, we have added: 

(p. 6, Supporting information): “For, example, it has been shown that although Democrats and Republicans in the US use similar moral terms when discussing political topics, they use them in different ways (1). Specifically, although these groups use words related to loyalty to a similar degree, Democrats use them to draw attention to the needs of underprivileged groups (e.g., “women”, “paid leave”), whereas Republicans use these words in the context of national security or religion.”

---

## [Decision Letter · Decision Letter 1]

26 Sep 2022

Two is better than one: Using a single emotion lexicon can lead to unreliable conclusions

PONE-D-22-09013R1

Dear Dr. Stillwell,

We’re pleased to inform you that your manuscript has been judged scientifically suitable for publication and will be formally accepted for publication once it meets all outstanding technical requirements.

Kind regards,

Andrea Fronzetti Colladon, Ph.D.

Academic Editor

PLOS ONE

Additional Editor Comments (optional):

I appreciate your revisions and believe the paper represents an excellent contribution to the journal.

Reviewers' comments:

Reviewer's Responses to Questions

**Comments to the Author**

1. If the authors have adequately addressed your comments raised in a previous round of review and you feel that this manuscript is now acceptable for publication, you may indicate that here to bypass the “Comments to the Author” section, enter your conflict of interest statement in the “Confidential to Editor” section, and submit your "Accept" recommendation.

Reviewer #2: All comments have been addressed

2. Is the manuscript technically sound, and do the data support the conclusions?

Reviewer #2: Yes

3. Has the statistical analysis been performed appropriately and rigorously? 

Reviewer #2: Yes

4. Have the authors made all data underlying the findings in their manuscript fully available?

Reviewer #2: Yes

5. Is the manuscript presented in an intelligible fashion and written in standard English?

Reviewer #2: Yes

6. Review Comments to the Author

Reviewer #2: The authors have worked hard to improve and refine the paper, and it shows. The revision addresses all of my earlier comments. The paper will be useful to a large number of people that use emotion lexicons. Congratulations on the fine work.

Final comment: Instead of citing the arXiv versions of papers, where applicable, cite the journal or conference proceedings where the papers were eventually published.

7. PLOS authors have the option to publish the peer review history of their article (what does this mean?). If published, this will include your full peer review and any attached files.

Reviewer #2: No

---

## [Editor Report · Acceptance letter]

5 Oct 2022

PONE-D-22-09013R1 

Two is better than one: Using a single emotion lexicon can lead to unreliable conclusions 

Dear Dr. Stillwell:

I'm pleased to inform you that your manuscript has been deemed suitable for publication in PLOS ONE. Congratulations! Your manuscript is now with our production department. 

Kind regards, 

on behalf of

Prof. Andrea Fronzetti Colladon 

Academic Editor

PLOS ONE